# Morphological diversity of single neurons in molecularly defined cell types

Hanchuan Peng[1,2,15 ✉], Peng Xie[2,15], Lijuan Liu[2,3,15], Xiuli Kuang[4], Yimin Wang[2,5], Lei Qu[2,6], Hui Gong[7,8], Shengdian Jiang[2], Anan Li[7,8], Zongcai Ruan[2], Liya Ding[2], Zizhen Yao[1], Chao Chen[4], Mengya Chen[5], Tanya L. Daigle[1], Rachel Dalley[1], Zhangcan Ding[2], Yanjun Duan[2], Aaron Feiner[1], Ping He[5], Chris Hill[1], Karla E. Hirokawa[1,14], Guodong Hong[2,3], Lei Huang[2], Sara Kebede[1], Hsien-Chi Kuo[1], Rachael Larsen[1], Phil Lesnar[1], Longfei Li[6], Qi Li[5], Xiangning Li[7,8], Yaoyao Li[4], Yuanyuan Li[6], An Liu[2,3], Donghuan Lu[9], Stephanie Mok[1], Lydia Ng[1], Thuc Nghi Nguyen[1,14], Qiang Ouyang[2], Jintao Pan[2], Elise Shen[1], Yuanyuan Song[2], Susan M. Sunkin[1], Bosiljka Tasic[1], Matthew B. Veldman[10], Wayne Wakeman[1], Wan Wan[6], Peng Wang[5], Quanxin Wang[1], Tao Wang[6], Yaping Wang[2], Feng Xiong[2], Wei Xiong[4], Wenjie Xu[1], Min Ye[4], Lulu Yin[2], Yang Yu[1], Jia Yuan[2,3], Jing Yuan[7,8], Zhixi Yun[2], Shaoqun Zeng[7], Shichen Zhang[2], Sujun Zhao[2], Zijun Zhao[2], Zhi Zhou[1], Z. Josh Huang[11,12], Luke Esposito[1], Michael J. Hawrylycz[1], Staci A. Sorensen[1], X. William Yang[10], Yefeng Zheng[9], Zhongze Gu[2], Wei Xie[2,3], Christof Koch[1], Qingming Luo[7,13], Julie A. Harris[1,14], Yun Wang[1] & Hongkui Zeng[1 ✉]

Dendritic and axonal morphology reflects the input and output of neurons and is a defining feature of neuronal types[1,2], yet our knowledge of its diversity remains limited. Here, to systematically examine complete single-neuron morphologies on a brain-wide scale, we established a pipeline encompassing sparse labelling, whole-brain imaging, reconstruction, registration and analysis. We fully reconstructed 1,741 neurons from cortex, claustrum, thalamus, striatum and other brain regions in mice. We identified 11 major projection neuron types with distinct morphological features and corresponding transcriptomic identities. Extensive projectional diversity was found within each of these major types, on the basis of which some types were clustered into more refined subtypes. This diversity follows a set of generalizable principles that govern long-range axonal projections at different levels, including molecular correspondence, divergent or convergent projection, axon termination pattern, regional specificity, topography, and individual cell variability. Although clear concordance with transcriptomic profiles is evident at the level of major projection type, fine-grained morphological diversity often does not readily correlate with transcriptomic subtypes derived from unsupervised clustering, highlighting the need for single-cell cross-modality studies. Overall, our study demonstrates the crucial need for quantitative description of complete single-cell anatomy in cell-type classification, as single-cell morphological diversity reveals a plethora of ways in which different cell types and their individual members may contribute to the configuration and function of their respective circuits.

Neurons exhibit extraordinary diversity across molecular, morphological, physiological and connectional features, thus accurate classification and mapping of cell types needs to consider and integrate these distinct yet related cellular properties[1,2]. Single-cell RNA-sequencing (scRNA-seq) has enabled systematic classification at the transcriptomic level[3–5], capturing major cell types with known anatomical and functional properties and revealing many potentially new cell types. Classification of cortical neurons using a combination of transcriptomic, electrophysiological and local morphological properties has also been achieved[6,7]. Brain-wide inter-areal connectivity has been mapped extensively using anterograde and retrograde tracing of projection neuron populations[8–11]. However, it remains largely unknown how population-level projection patterns are reflected at the level of single cells, the fundamental units of the circuits. Thus, characterizing single neuron axonal projections through reconstruction of complete morphologies provides ground-truth information not only for cell classification, but also for charting global networks and local circuits.

Single neurons have traditionally been labelled with molecular markers using whole-cell patching, in vivo electroporation[12–14], sparse transgenic expression[15] or sparse viral infection[16–19], followed by manual reconstruction across many consecutive sections. The recent development of high-throughput and high-resolution fluorescent imaging platforms, such

as fluorescence micro-optical sectioning tomography (fMOST)[20,21] and MouseLight[22,23], has enabled the generation of large-scale datasets for neuron reconstructions. Further improvements in brain-wide single-cell labelling methods and computational tools to expedite the laborious reconstruction process are still needed to achieve scalable and complete reconstructions from a comprehensive set of cell types.

As part of the BRAIN Initiative Cell Census Network (BICCN) efforts to characterize brain cell types across multiple modalities, we established a pipeline to label, image, reconstruct and classify single neurons in mice. We report here the largest set of complete single-neuron reconstructions to date. These neurons are labelled by cell subclass or type-selective Cre driver lines, enabling correlation of their morphologies and projection patterns with molecular identities. We also provide a corresponding set of scRNA-seq data from retrogradely labelled neurons (Retro-seq data) to corroborate our anatomical findings. Overall, our study reveals substantial morphological and projection diversity of individual neurons—this diversity is governed by underlying principles that manifest in region- and cell-type-specific manners.

## Results

### Complete neuron reconstruction pipeline

To achieve more efficient, widespread, consistently sparse yet strong labelling, we used TIGRE2.0 transgenic reporter lines that exhibit viral-like transgene-expression levels, coupling them with Cre expression from either driver lines or viral delivery. We used two types of reporter lines: the GFP-expressing Ai139 or Ai140 TIGRE2.0 reporter[24], optionally coupled with the Ai82 TIGRE1.0 reporter[25]; and the TIGRE-MORF reporter[26] (also called Ai166) (Extended Data Fig. 1). TIGRE-MORF (Ai166) expresses the MORF gene, which is a farnesylated eGFP (GFPf) preceded by a mononucleotide repeat of 22 guanines ($G_{22}$–GFPf). The GFPf transgene is translated only when rare stochastic frameshift events occur to delete one guanine, leading to extremely sparse labelling well suited for reconstruction of elaborate axonal arborizations. For this study, we generated 53 high-quality fMOST-imaged brain datasets with sparsely labelled cells in cortical, thalamic, claustral, striatal and other regions, and for cholinergic, noradrenergic and serotonergic neuron types (Extended Data Fig. 2, Supplementary Table 1).

We acquired whole-brain images with sufficient resolution for reconstructing fine-calibre axons using the fMOST imaging platform[21]. To handle the large imaging datasets generated, we established a standardized image data processing and informatics workflow (Extended Data Fig. 3) for efficient whole-brain morphology reconstruction using Vaa3D, an open-source, cross-platform visualization, reconstruction and analysis system[27,28]. In parallel, each fMOST dataset was registered to the 3D Allen mouse brain Common Coordinate Framework (CCFv3)[29], using a newly developed mBrainAligner program specifically designed for fMOST datasets to handle the challenges of brain shrinkage and deformation (Extended Data Fig. 4a). Following registration of the whole-brain image dataset, all individual neuron reconstructions were also registered to CCFv3 using the source brain's transformation parameters. Co-registration of reconstructions from different brains to CCFv3 enables digital anatomical delineation, spatial quantification and comparison of each reconstructed morphology and its compartments (for example, soma, dendrites and axon arbors) using a set of analysis tools that we developed. A stringent quality control process was established to ensure the completeness of reconstructed morphologies (Extended Data Fig. 4b–e). An advantage of this platform is the distributed and modularized components and open access of all data and tools that facilitate multi-site collaboration and community engagement.

### Overview of projection neuron types

To extract rules underlying the morphological diversity of long-range projection neurons, we systematically analysed eight subclasses of neurons that were labelled by Cre lines representing specific transcriptomic subclasses[4,30,31] (Fig. 1, Supplementary Table 2).

From the *Cux2-CreERT2;Ai166* mice which label cortical layer (L)2/3 and L4 intratelencephalic (IT) subclasses, we analysed 100 neurons in somatosensory and motor regions: primary somatosensory cortex (SSp, $n = 46$), supplemental somatosensory cortex (SSs, $n = 14$), primary motor cortex (MOp, $n = 22$) and secondary motor cortex (MOs, $n = 18$). From the *Plxnd1-CreER;Ai166* mice, in which cortical L2/3 and L5 IT subclasses are labelled, we analysed 33 L5 IT neurons in SSp ($n = 15$), SSs ($n = 10$), MOp ($n = 4$) and MOs ($n = 4$). From the *Fezf2-CreER;Ai166* and *Pvalb-T2A-CreERT2;Ai166* mice, in which the cortical L5 extratelencephalic (ET) (also known as pyramidal tract (PT)) subclass is labelled, we analysed 197 neurons in SSp ($n = 141$), SSs ($n = 21$), MOp ($n = 19$) and MOs ($n = 16$).

We investigated a special type of cortical excitatory neurons, the *Car3* IT transcriptomic subclass[4,31], whose morphology and projection pattern were unknown. This subclass of neurons is located in the deep layers (mostly L6) of all lateral cortical areas and shares the same transcriptomic clusters with neurons from the claustrum (CLA). Mesoscale population anterograde tracing shows that cortical L6 *Car3* neurons have a more restricted intracortical projection pattern than CLA neurons, which project widely in cortex[32] (Extended Data Fig. 5). We analysed 99 neurons from the *Gnb4-IRES2-CreERT2;Ai140;Ai82* mice, including 29 CLA neurons and 70 neurons from multiple lateral cortical areas.

We analysed 701 thalamocortical projection neurons from the *Tnnt1-IRES-CreERT2;Ai140;Ai82* and *Vipr2-IRES2-Cre-neo;Ai166* mice, in which the *Prkcd_Grin2c* transcriptomic subclass is labelled (H.Z. et al., unpublished data). The reconstructed neurons cover 21 of the 44 thalamic regions in CCFv3, which can be broadly divided into two major groups[33,34], the 'core' or 'driver' nuclei ($n = 638$ cells) and the 'matrix' or 'modulatory' nuclei ($n = 63$ cells).

We analysed 280 striatal (caudoputamen (CP)) neurons from the *Tnnt1*, *Vipr2* and *Plxnd1* Cre lines. These are the medium spiny neurons (MSNs) with main projections to either globus pallidus, external segment (GPe, $n = 180$ cells) or substantia nigra, reticular part (SNr, $n = 100$ cells), which correspond to the two well-known subclasses of MSNs, dopamine receptor D1 (*Drd1*) neurons projecting to SNr (direct pathway) and dopamine receptor D2 (*Drd2*) neurons projecting to GPe (indirect pathway)[35].

To provide a clearer narrative, here we first summarize major findings derived from the detailed characterizations described in the sections below. We analysed morphological features and rules at multiple levels: projection class, projection type, projection patterns (such as convergent or divergent projection, feedforward or feedback projection, and total number of projection targets), regional difference, topography and individual cell variability (Fig. 1b).

At the higher levels, neurons from the 8 transcriptomic subclasses exhibit highly distinct projection patterns and correspond well to 5 projection classes and 11 projection types. The split of the L5 ET subclass into medulla (MY)-projecting and non-MY-projecting types corresponds to specific transcriptomic and epigenomic types within the L5 ET subclass described in other studies[4,36,37]. The split of the thalamic *Prkcd_Grin2c* subclass into core and matrix projection types is also consistent with transcriptomic clusters differentiating these thalamic nuclei (H.Z. et al., unpublished data; see also ref. [38]). By contrast, the split of the *Car3* subclass into L6 *Car3* and CLA projection types is not associated with corresponding transcriptomic clusters.

Beyond these high-level divisions, morphological distinctions among the 11 projection types are reflected in multiple aspects. The average number of projection targets (each target is defined by having total axon length greater than 1 mm)[12] is highly characteristic of each projection type, with *Car3* (in particular CLA) neurons having the highest number of targets, followed by L5 ET neurons, then by IT and thalamic matrix neurons, and thalamic core and CP neurons having the lowest. This distinction appears to be directly related to the differential

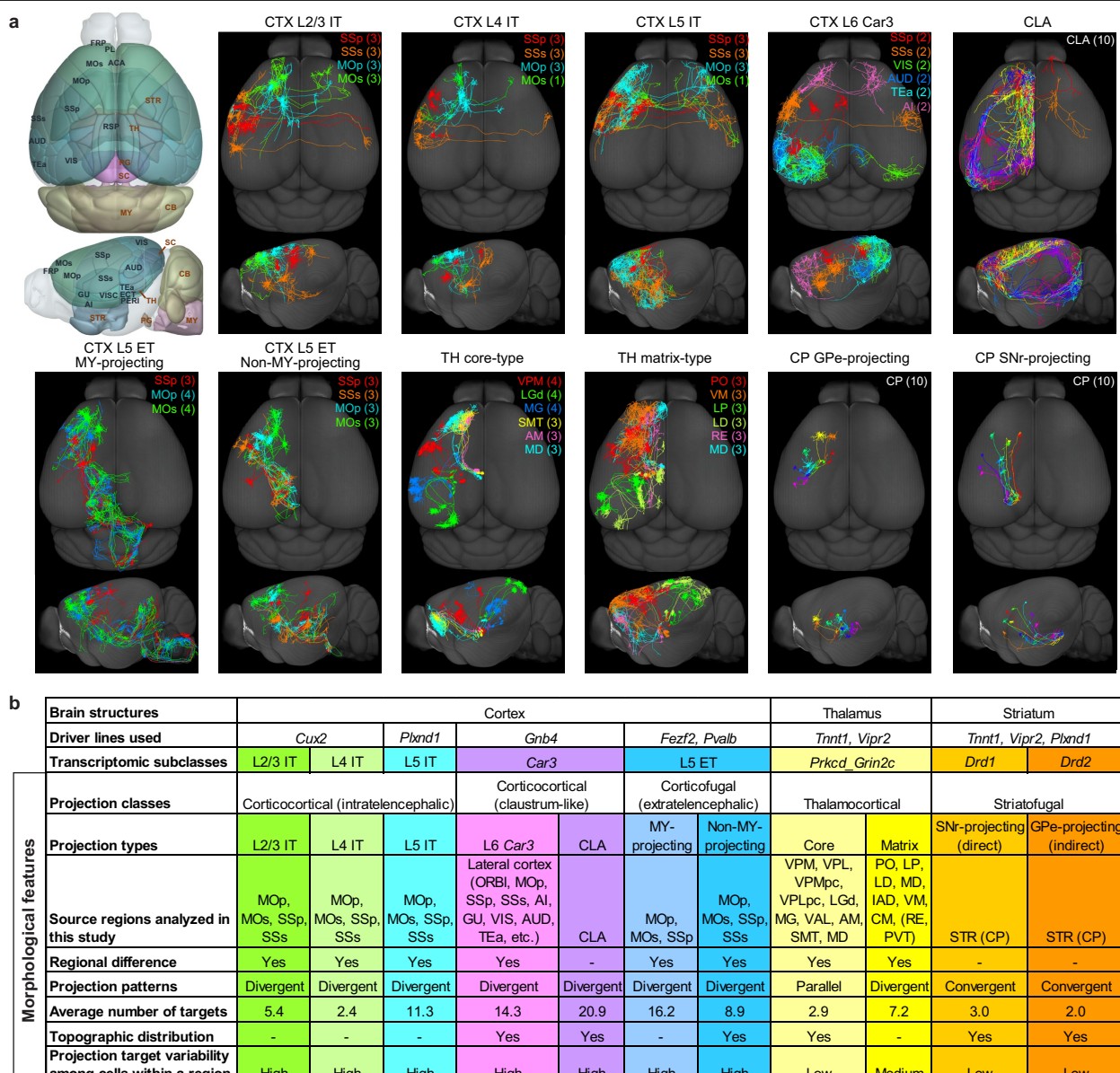

**Fig. 1 | Morphological and projectional properties of 11 long-range projection neuron types at the single-cell level. a**, Example single-neuron morphologies for each of the 11 projection neuron types. Numbers in parentheses denote the number of neurons shown in each indicated region. In this and all subsequent figures, neurons are flipped to the left hemisphere for comparison of axon projection patterns. Left, CCFv3 3D brain models with anatomical delineation of all cortical and selected subcortical regions (striatum (STR), TH, superior colliculus (SC), PG, MY and cerebellum (CB)). **b**, Summary of the projection neuron types and their morphological and projectional features. Hyphens denote features not investigated in this study.

Our transcriptomic study (H.Z. et al., unpublished results) suggests that most of these thalamocortical projection neurons are in the *Prkcd_Grin2c* transcriptomic subclass, whereas those from nucleus of reuniens (RE) and paraventricular nucleus (PVT) are not. ACA, anterior singulate area; AI, agranular insular area; AM, anteromedial nucleus; AUD, auditory areas; CM, central medial nucleus; GU, gustatory area; IAD, interanterodorsal nucleus; LD, lateral dorsal nucleus; RSP, retrosplenial area; SMT, submedial nucleus; VIS, visual area; VISC, visceral area; VM, ventral medial nucleus; VPMpc, ventral posteromedial nucleus, parvicellular part.

projection patterns (divergent, parallel or convergent) among types. A major distinction between the two corticocortical projecting classes is that the CLA and cortical L6 *Car3* neurons do not have collateral projections into striatum whereas IT neurons do. In addition, multiple morphological features distinguish among L2/3, L4 and L5 IT types, and between thalamic core and matrix types—some of these features are likely to be associated with the differential roles of these neuron types in mediating feedforward or feedback information flow.

We observe further morphological diversity within each projection type. Regional specificity is seen in all cortical and thalamic projection types containing neurons from different subregions. Topographic

correspondence between soma locations and major axon arbors is seen in all projection types with sufficient numbers of reconstructions that allow such examination. Within each projection type that has divergent projection patterns, there is a high degree of variability among individual cells in choosing a subset of projection targets. Such individual variability appears stochastic, while it also allows further clustering of individual cells into target-driven subtypes as we have done in the most highly divergent *Car3* and L5 ET projection types.

To directly compare the single-cell and population-level projection patterns, we identified 1,354 single-cell morphologies and 163 mesoscale experiments from the Allen Mouse Brain Connectivity Atlas,

matched on the basis of soma or injection site being within the same CCFv3 structure (Supplementary Table 3). In this location-matched dataset, the combined single-cell projection pattern from a given region (and cortical layer) is highly concordant with that of the mesoscale experiments, with a few exceptions (Extended Data Fig. 6). A minimal set of single cells can recapitulate the mesoscale pattern well; however, averaging across all single cells shows a low level of approximation, suggesting highly diverse projection patterns among the single cells.

Overall, this large set of long-range projection neurons displays a wide range of morphological and projectional diversity that can be described at multiple levels, revealing different rules that different neuron types follow. Morphological features at high levels are closely related to the neuron's molecular identities, whereas those at more refined levels may underlie the specific functional role of each neuron in the circuit it is embedded in.

### Cortical L2/3, L4 and L5 IT neurons

All the cortical L2/3, L4 and L5 IT neurons have their long-range projections confined within cortex and striatum (Extended Data Fig. 7). We compared single neuron and population long-range projections for both L2/3 and L5 IT subclasses (Fig. 2a). All neurons within a type collectively recapitulate the population projection pattern, but each neuron selects a subset of projection targets. Such selection appears random without correlation to each neuron's soma depth or dendritic morphology. L5 IT neurons have significantly greater numbers of projection targets than L2/3 neurons, and this difference is particularly pronounced in SSp and SSs (Fig. 2a,b). L5 IT neurons also have longer total axon lengths in ipsilateral and contralateral cortex and striatum (Fig. 2b). L4 neurons exhibit a notable regional difference; all but two SSp L4 neurons have only local axons but no long-range projections, whereas nearly all L4 neurons in SSs, MOp and MOs do have axon projections outside of their local area (Extended Data Fig. 7).

We examined both local and distal axon projections to see whether axon termination patterns of single cells recapitulate the overall feedforward or feedback projection patterns apparent at the population level[33]. Quantitative vertical profiles of local axons (Fig. 2c) show that, in addition to their within-layer collaterals, L2/3 cells also project downward into L5, whereas L5 IT cells have local projections up to L1. This difference is consistent across all cortical areas examined.

Given that each IT neuron projects to only a subset of their potential intracortical targets, we pooled all the distal axon arbors from all the cells within the L2/3 or L5 IT subclass and within a source region to generate a cumulative vertical profile of laminar distribution pattern (Fig. 2d, Extended Data Fig. 8). In SSp and SSs, distal axon terminals of L2/3 cells are concentrated in middle layers (L2/3-5), whereas those of L5 IT cells preferentially target L1, suggesting feedforward and feedback roles for these L2/3 and L5 IT cells, respectively. Notably, in MOp and MOs, distal axon terminals of both L2/3 and L5 IT cells preferentially target L1 and to a lesser extent L2/3, suggesting that L2/3 IT cells in motor cortex may also have a feedback role onto other cortical regions. This cell-type difference between somatosensory and motor cortices is consistent with the differential positions of these regions in the hierarchical cortical network[33] (that is, MOp and MOs are higher than SSp and SSs).

Overall, these analyses reveal major projectional differences both among the L2/3, L4 and L5 IT subclasses and among the cortical regions, as well as individual cell-to-cell variations within each subclass and each region.

We also investigated the projection target specificity of transcriptomic cell types using Retro-seq[4], in which the transcriptomes of 1,134 retrogradely labelled neurons from SSp, SSs, MOp and MOs were mapped to our established transcriptomic taxonomy across the entire isocortex and hippocampal formation[31] (Extended Data Fig. 9, Supplementary Table 4). For each source region, within each of the L2/3, L4, and L5 IT subclasses, neurons labelled from different injection targets were mostly mapped to a common subset of transcriptomic

types, with little between-target difference, suggesting that within each IT subclass, projection pattern at single-cell level does not correlate one-to-one with the cell's transcriptomic type.

### Cortical L5 ET neurons

L5 ET neurons exhibit extensive heterogeneity in their selected subset of projection targets (Supplementary Fig. 1). To search for patterns in this diversity, we clustered all L5 ET neurons ($n = 193$) together and found that cluster segregation is mainly driven by projection targets in thalamus (TH), midbrain (MB) and MY (Fig. 3). A major division is between neurons mainly projecting to MY and MB structures such as midbrain reticular nucleus (MRN) and superior colliculus, motor related (SCm) (clusters 5–6) and neurons mainly projecting to TH (the other clusters). The MY- and MRN-projecting neurons are further subdivided into those preferentially projecting to zona incerta (ZI) (cluster 5) or SCm (cluster 6). The TH branch is subdivided into those mainly projecting to ventral anterior-lateral complex (VAL) (cluster 1), posterior complex (PO) (clusters 2–4) or ventral posteromedial nucleus (VPM) and ventral posterolateral nucleus (VPL) (clusters 7–8). In addition to PO, cluster 3 neurons also project to mediodorsal nucleus (MD), central lateral nucleus (CL) and parafascicular nucleus (PF). Other brain regions such as CP, GPe, SNr and pontine grey (PG) are shared projection targets among all or most L5 ET neurons.

L5 ET neurons belonging to different clusters are intermingled in the cortical regions they come from (Extended Data Fig. 10a). At the same time, most MOp and MOs neurons are found in the more complex, MY and MRN-projecting clusters 5 and 6, whereas the other clusters with simpler, TH projections mainly contain SSp and SSs neurons, revealing a regional difference and suggesting that medulla projection may be primarily a feature of MOp and MOs neurons (Fig. 3a).

### Cortical and claustral *Car3* neurons

All CLA and cortical (CTX) L6 *Car3* neurons project extensively in cortex, but they do not project into striatum. Clustering on the 99 *Car3* neurons from all regions identified 13 clusters (Fig. 4a, b). The first major division is between CLA and L6 *Car3* neurons. CLA neurons have greater total axon lengths and higher numbers of projection targets (Fig. 4c).

CLA neurons often have long-distance projections, predominantly targeting prefrontal, medial and lateral association cortical areas as well as entorhinal cortex, whereas CTX L6 *Car3* neurons mostly project to nearby cortical areas or homotypic cortical areas on the contralateral side (Fig. 4b, d). A prominent observation is that both CLA and CTX L6 *Car3* clusters are arranged topographically from anterior to posterior parts, based on both soma location and projection targets—each cluster contains a group of neurons that are located close to each other and project to similar cortical target areas (Fig. 4d).

In our single-cell transcriptomic taxonomy of the mouse cortex and hippocampus[31], the *Car3* subclass is highly distinct from other glutamatergic neuron subclasses (Extended Data Fig. 11a). Single-cell transcriptomes of CLA neurons also mapped exclusively to this subclass. We performed Retro-seq on cells isolated from CLA ($n = 240$) and cortical areas SSs ($n = 11$) and temporal association area (TEa)–perirhinal area (PERI)–ectorhinal area (ECT) ($n = 35$) that were labelled by retrograde tracers injected into far-apart cortical areas (Supplementary Table 4). The Retro-seq cells mapped to the *Car3* subclass are concentrated in one of the three clusters (Extended Data Fig. 11b). These results suggest a lack of regional distinction and that these cortical and claustral cells are highly related to each other, possibly having a common developmental origin. In an attempt to reveal subtler transcriptomic differences, we re-clustered all the non-Retro-seq *Car3* cells ($n = 1,699$) from cortex and CLA (Supplementary Table 4), resulting in 8 clusters, and then remapped all the Retro-seq cells to the new clusters (Extended Data Fig. 11c, d). The CLA and CTX L6 *Car3* Retro-seq cells projecting to different cortical areas are again distributed across a similar set of clusters, indicating no clear one-to-one correspondence between transcriptomic clusters and projection target specificity.

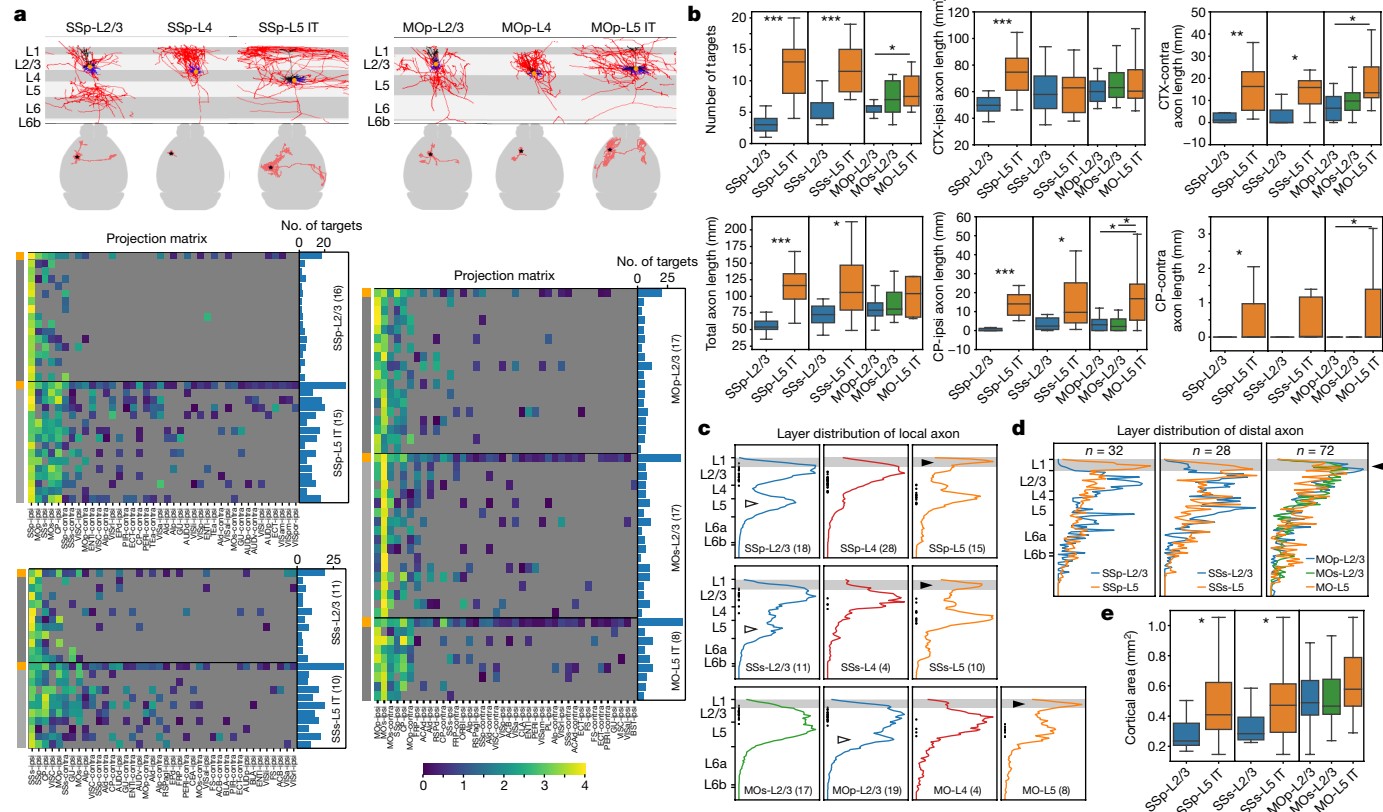

**Fig. 2 | Local morphology and long-range projection of cortical L2/3, L4 and L5 IT neurons. a**, Projection matrices comparing long-range projection patterns between individual neurons and mesoscale population-level projections, and between L2/3 and L5 IT neurons in each cortical region. Example SSp and MOp neurons are shown above the matrices, with their local morphologies (top row; apical dendrite in black, basal dendrite in blue, axon in red and soma as an orange dot) and intracortical long-range projections (bottom row; axon in red and soma as a star). The first row of each matrix (labelled orange in the side bar) shows the mesoscale projection pattern for each cell type and region, collapsed from multiple mesoscale experiments. Each of the subsequent rows shows the projection pattern for a single neuron. Bar graphs to the right show the number of projection target regions for each mesoscale or single cell. MOs and MOp L5 IT neurons are grouped together owing to low numbers in each region. Heat map colours represent projection strengths, defined as ln(NPV × 100 + 1) for mesoscale experiments and ln(axon length) for single cells, where NPV is normalized projection volume. Target regions are defined using thresholds of ln(NPV × 100 + 1) > 0.2 for mesoscale experiments and axon length > 1 mm for single cells. Regions below the thresholds are shown in grey. The same definitions are used for all other figures. **b**, Comparison of numbers of targets and axon lengths between L2/3 and L5 IT neurons across different regions. In all figures, box edges in box plots show 25th and 75th percentiles, the centre line shows the 50th percentile, and bars show 1.5× the interquartile range (75th percentile – 25th percentile). **c**, Comparison of vertical profiles of local axon projections among L2/3, L4 and L5

IT neurons. Vertical profiles are combined from all neurons in each type and region (with numbers of cells in parentheses). Soma locations are indicated as dots along the left edge of each plot. Black and white arrowheads point to axon projection differences observed in L1 and L5, respectively. **d**, Comparison of cumulative vertical profiles of distal axon projections in target cortical regions between L2/3 and L5 IT neurons across different source regions. The black arrowhead points to the axon projection difference observed in L1. **e**, Comparison of the tangential span of distal axon projections in target cortical regions between L2/3 and L5 IT neurons across different source regions. Cell numbers in parentheses in **a** are used for quantifications in **b**, **e**. *P < 0.05, **P < 0.001, ***P < 0.0001, two-sided Mann-Whitney U test, without adjustment for multiple comparison. ACAd, anterior cingulate area, dorsal part; ACB, nucleus accumbens; AId, agranular insular area, dorsal part; AIp, agranular insular area, posterior part; AUDd, dorsal auditory area; AUDp, primary auditory area; AUDv, ventral auditory area; BLA, basolateral amygdalar nucleus; BST, bed nuclei of the stria terminalis; CEA, central amygdalar nucleus; ENTl, entorhinal area, lateral part; EPd, endopiriform nucleus, dorsal part; FRP, frontal pole; FS, fundus of striatum; ORBl, orbital area, lateral part; PIR, piriform area; PL, prelimbic area; RSPagl, retrosplenial area, lateral agranular part; RSPd, retrosplenial area, dorsal part; VISa, anterior visual area; VISal, anterolateral visual area; VISam, anteromedial visual area; VISl, lateral visual area; VISli, laterointermediate visual area; VISp, primary visual area; VISpm, posteromedial visual area; VISpor, postrhinal area; VISrl, rostrolateral visual area; contra, contralateral; ipsi, ipsilateral.

## Thalamic core and matrix neurons

Single thalamic sensory–motor relay neurons usually have one major axon arbor targeting their corresponding primary sensory or motor cortex. Axons from these nuclei terminate predominantly in L4, consistent with the core-type classification (Extended Data Fig. 12a–e, Supplementary Fig. 2). A small fraction of the core-type thalamic neurons have more than one axon arbor targeting different cortical areas. We analysed morphometric features of 944 axon arbors from 586 neurons located in the sensory thalamic nuclei VPM, VPL, lateral geniculate complex, dorsal part (LGd) and medial geniculate complex (MG) (Extended Data Fig. 13). We identify two major types of axon arbors targeting cortical layer 4

(and lower L2/3), a smaller type 1 and a larger type 2. Thus, core-type neurons can be assigned to either small-arbor or large-arbor subtype.

Outside the sensory–motor relay nuclei, nearly all reconstructed thalamic neurons have a large, diffusely branched axon arbor and/or several arbors projecting to different cortical areas, often with columnar or L5-dominant axon termination patterns. Many (79%) of these cells also have axon branches more than 1 mm long in L1, consistent with the matrix type, but they also exhibit a diverse range of morphological patterns (Extended Data Fig. 12e–h, Supplementary Fig. 3). Some nuclei (for example, lateral posterior nucleus (LP) and mediodorsal nucleus (MD)) further display distinct subdivisions.

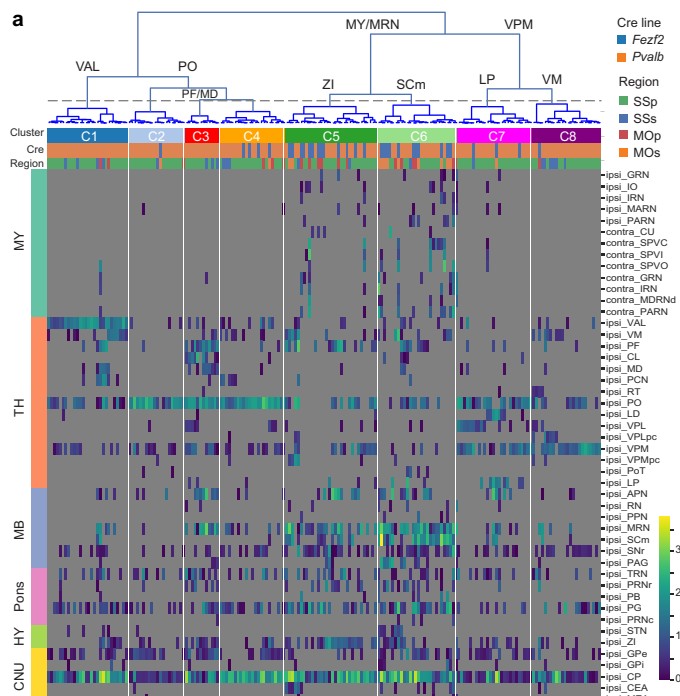

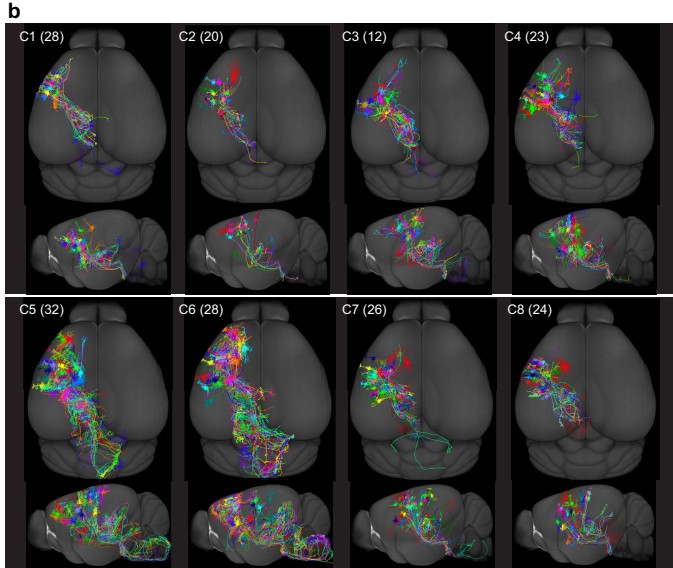

**Fig. 3 | Long-range projection of cortical L5 ET neurons. a**, Clustering based on long-range projection targets of L5 ET neurons from all regions (MOs, MOp, SSp and SSs) combined, and projection matrix heat map organized by clustering result. Main target regions driving each cluster division are shown on the dendrogram. The dashed line indicates the threshold for cluster calls. For the projection matrix, columns represent single cells and rows represent target regions. Heat map colours represent projection strengths, defined as ln(axon length). **b**, Whole-brain-projection overview (top–down and side views) of individual neurons in each cluster (all cells shown for each cluster, numbers of cells in parentheses). APN, anterior pretectal nucleus; CNU, cerebral nuclei; CU, cuneate nucleus; GRN, gigantocellular reticular nucleus; HY, hypothalamus; IO, inferior olivary complex; IRN, intermediate reticular nucleus; MARN, magnocellular reticular nucleus; MDRNd, medullary reticular nucleus, dorsal part; MEA, medial amygdalar nucleus; PAG, periaqueductal grey; PARN, parvicellular reticular nucleus; PB, parabrachial nucleus; PCN, paracentral nucleus; PoT, posterior triangular thalamic nucleus; PPN, pedunculopontine nucleus; PRNc, pontine reticular nucleus, caudal part; PRNr, pontine reticular nucleus; RN, red nucleus; RT, reticular nucleus; SPVC, spinal nucleus of the trigeminal, caudal part; SPVI, spinal nucleus of the trigeminal, interpolar part; SPVO, spinal nucleus of the trigeminal, oral part; STN, subthalamic nucleus; TRN, tegmental reticular nucleus; VPLpc, ventral posterolateral nucleus, parvicellular part; VPMpc, ventral posteromedial nucleus, parvicellular part.

arbors decreases exponentially, along with the increase of the pair's soma-to-soma distance (Extended Data Fig. 14c), indicating a regular spatial organization of these neurons' axon projections. The axon arbor distances between striatal neurons within the same type are substantially smaller and their overlap scores are substantially higher than those of neurons from various thalamic nuclei (Extended Data Fig. 14d). Furthermore, axon arbors in GPe or SNr can be grouped into domains on the basis of the degree of overlap; these domains are arranged topographically and correspond to the topographic localization of the somas in striatum (Extended Data Fig. 14e).

### Topographic organization of axon projection

Single-neuron reconstructions enabled us to investigate the topographic relationship between soma locations and main axon arbors of several projection neuron types for which we have sufficiently large numbers of reconstructions, and soma–axon arbor spatial topographic relationships were found in all cases examined (Extended Data Fig. 15). For neurons in LGd, VPM, and VPM and VPL combined, axon arbor positions along anterior–posterior and lateral–medial dimensions in each of the cortical target areas correspond to soma positions along ventral–dorsal and medial–lateral dimensions in each thalamic nucleus, indicating a three-dimensional rotation of axon projections (Extended Data Fig. 15a–c). This topographic relationship is most clearly seen in VPM and VPL neurons, whereas it is more complex for LGd neurons. There is a similar topographic rotational relationship for SSp L5 ET neurons between their soma locations in SSp and axon arbors in VPM and PO, although the relationship also appears fuzzier (Extended Data Fig. 15d). For striatal neurons, GPe-projecting MSNs maintain the dorsal–ventral, lateral–medial and anterior–posterior orientations between their somas in CP and axon arbors in GPe, whereas SNr-projecting MSNs exhibit a flip between their soma and axon arbor positions in the dorsal–ventral axis (Extended Data Fig. 15e, f), consistent with previous bulk tracing studies[39]. These single-cell results reveal yet another level of organization of cell-type-specific axon projection patterns.

### Discussion

To fully understand the anatomical diversity and specificity of individual neurons across the mammalian brain, a very large number of neurons will need to be examined. Approaches such as MAPseq and BARseq[12,40,41] can quickly survey projection patterns at regional level for many neurons in a high-throughput manner. However, many essential details can

A quantitative interareal projection matrix (Extended Data Fig. 12i, Supplementary Tables 2, 3) further demonstrates the distinction between core- and matrix-type neurons, with the core-type neurons predominantly targeting a single cortical area (sometimes with a secondary area) and the matrix-type neurons targeting multiple cortical areas. Within each nucleus, individual neurons show a high (for core cells) or moderate (for matrix cells) degree of consistency with each other and with the population projection pattern for that nucleus.

### Striatal medium spiny neurons

Individual striatal MSNs project to GPe or SNr in a simple point-to-point fashion, each with one major axon arbor. Most SNr-projecting neurons also send minor collaterals to globus pallidus, internal segment (GPi) and/or GPe. The dominant feature of both types of striatal neurons is convergent projection within the main target region, GPe or SNr, consistent with the approximately 20-fold smaller sizes of these regions compared to the dorsal striatum (Extended Data Fig. 14a–c). Between each pair of neurons, the centre-to-centre distance of their axon arbors increases proportionally and the degree of overlap between the axon

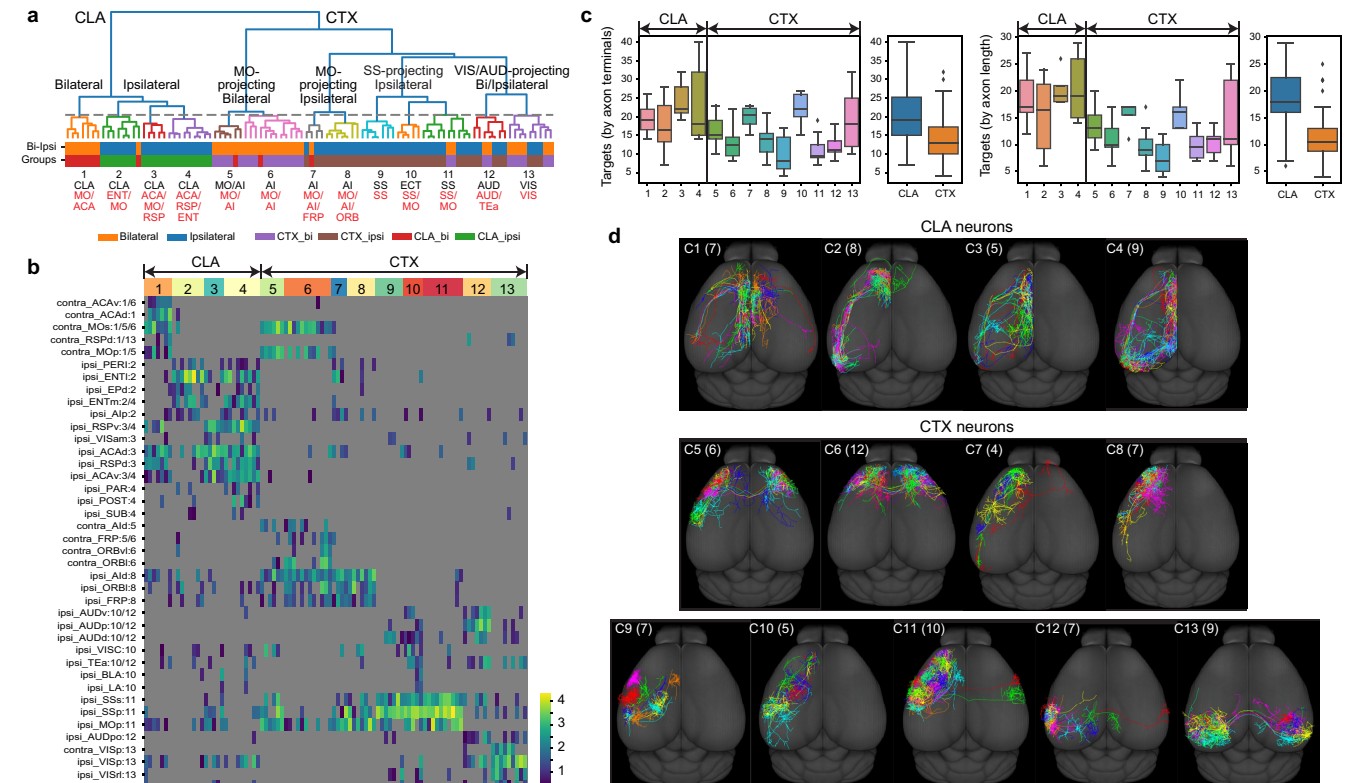

**Fig. 4 | Projection diversity of cortical and claustral *Car3* neurons.**
**a**, Clustering of *Car3* neurons from all regions based on four feature sets: projection pattern, soma location, axon morphology and dendrite morphology. The dashed line indicates the threshold for cluster calls. Only clusters with a minimum of three cells are shown; thus, three cortical cells are omitted. Each cluster is annotated by the main brain regions where somas (black) and axon terminals (red) reside. Regions are selected to represent more than 50% of cluster members. Bi-ipsi, bilateral or ipsilateral. **b**, Projection matrix with cells sorted by cluster assignment. Columns represent single cells. Rows represent targets, and the number following each target name indicates the dominant cluster ID for the row. Heat map colours represent projection strengths, defined as ln(axon length). We identified four CTX L6 *Car3* cells from ECT and several CLA neurons with axon collaterals projecting into amygdala areas, consistent

with previous studies[32,46]. **c**, Total number of cortical targets innervated by each neuron grouped by clusters. Two different thresholds are used to label a region as targeted. With a threshold of at least one terminal bouton, we find an average of 18 targets for CLA and 11 for CTX *Car3* neurons. Using a minimum of 1 mm of axon length results in 21 and 14 targets for CLA and CTX *Car3* neurons, respectively. Cell numbers are shown in **d**. Whiskers show outliers below minima or above maxima. **d**, Whole-brain top-down view of neurons in each cluster (all cells are shown for each cluster, with cell number in parentheses). ACAd, anterior cingulate area, dorsal part; ACAv, anterior cingulate area, ventral part; ENT, entorhinal area; ENTm, entorhinal area, medial part; LA, lateral amygdalar nucleus; MO, motor cortex; ORB, orbital area; ORBvl, orbital area, ventrolateral part; PAR, parasubiculum; POST, postsubiculum; RSPv, retrosplenial area, ventral part; SS, somatosensory cortex; SUB, subiculum.

be obtained only through complete morphological reconstructions. Collecting such ground-truth data provides a unique opportunity to uncover principles of neuronal diversity and circuit organization.

We systematically examined multiple levels of morphological properties in this large set of complete reconstructions with the goal of deriving organizational rules governing long-range axon projections, incorporating cross-modality relationship between transcriptomic and morphological properties. At the highest level, there is a high degree of concordance between major transcriptomic and projection neuron types. Neurons belonging to different transcriptomic subclasses have highly distinct morphological and projectional properties. Additionally, the medulla-projecting and non-medulla-projecting L5 ET neuron types and the core and matrix thalamocortical projection types also correlate with different molecular types as shown in other studies[36–38]. An exception at this level is the claustral and cortical L6 *Car3* neurons which have distinct projection patterns but appear transcriptomically homogeneous.

At the intermediate level, within each projection neuron type, neurons follow region-specific and topographic organizational rules. Our latest transcriptomic study showed that L2/3 IT, L4 IT, L5 IT and L5 ET transcriptomic types are largely shared between somatosensory and motor cortical areas with some continuous gradient variations, and the *Car3* transcriptomic types are also shared among all lateral cortical areas and claustrum[31]. By contrast, here we show for all types that, within each type,

neurons from different regions have distinct sets of projection targets that are region-specific. Furthermore, in each type and region examined we observe a topographic relationship between soma locations and axon arbor distributions in a main target region. The most prominent example is the cortical and claustral *Car3* neurons whose extensive variation of axon projections is linked to both regional specificity and topography, two closely related factors as these neurons are situated at the lateral part of the cortical sheet along almost the entire anterior–posterior extent.

At the lowest, single-cell level, the degree of similarity or variability between individual neurons within a given type also varies across types. Within-type individual cell variability is high in cortical and claustral neurons, moderate in thalamic matrix neurons, and low in thalamic core neurons and striatal MSNs.

A major question is how morphological and projectional properties compare and correlate with the neurons' molecular identities. We attempted to address this question with two approaches: using validated driver lines to define the subclass-level molecular identities of reconstructed neurons and using Retro-seq to obtain transcriptomic profiles of neurons projecting to specific targets. Both approaches show that subclasses of neurons have highly distinct morphological and projectional properties; however, within these major types, especially for cortical and claustral neurons, many aspects of morphological diversity cannot be accounted for by currently identified transcriptomic

subtypes or clusters in the adult stage. Previous studies showed that L2/3 SSp pyramidal neurons projecting to MOp or SSs have distinct intrinsic and network physiological properties[42,43]. Even though they may not belong to distinct transcriptomic subtypes, it will be interesting to examine gene-expression differences that might correspond to the differential connectional and physiological properties for these neurons, as found for primary visual cortical neurons projecting differentially to medial or lateral higher visual areas[44].

Several mechanisms may explain the origin of the morphological diversity, such as molecular instructions that act transiently during development[45], activity-dependent cell interactions, or stochastic processes. It will be informative to develop methods that enable complete reconstruction of morphology and in-depth gene-expression profiling to be conducted on the same cell, and apply them to single cells in both adult stage and during brain development, so that the developmental correlations of molecular and morphological and connectional features can be identified.

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

[1]Allen Institute for Brain Science, Seattle, WA, USA. [2]SEU-ALLEN Joint Center, Institute for Brain and Intelligence, Southeast University, Nanjing, China. [3]Ministry of Education Key Laboratory of Developmental Genes and Human Disease, School of Life Science and Technology, Southeast University, Nanjing, China. [4]School of Optometry and Ophthalmology, Wenzhou Medical University, Wenzhou, China. [5]School of Computer Engineering and Science, Shanghai University, Shanghai, China. [6]Key Laboratory of Intelligent Computation and Signal Processing, Ministry of Education, Anhui University, Hefei, China. [7]Britton Chance Center for Biomedical Photonics, Wuhan National Laboratory for Optoelectronics, MoE Key Laboratory for Biomedical Photonics, Huazhong University of Science and Technology, Wuhan, China. [8]HUST-Suzhou Institute for Brainsmatics, JITRI Institute for Brainsmatics, Suzhou, China. [9]Tencent Jarvis Lab, Shenzhen, China. [10]Center for Neurobehavioral Genetics, Jane and Terry Semel Institute for Neuroscience and Human Behavior, Department of Psychiatry and Biobehavioral Sciences, David Geffen School of Medicine, University of California, Los Angeles, Los Angeles, CA, USA. [11]Cold Spring Harbor Laboratory, Cold Spring Harbor, NY, USA. [12]Department of Neurobiology, Duke University School of Medicine, Durham, NC, USA. [13]School of Biomedical Engineering, Hainan University, Haikou, China. [14]Present address: Cajal Neuroscience, Seattle, WA, USA. [15]These authors contributed equally: Hanchuan Peng, Peng Xie, Lijuan Liu. ✉e-mail: h@braintell.org; hongkuiz@alleninstitute.org

## Methods

### Nomenclature and abbreviations in CCFv3 ontology of mouse brain regions mentioned in this study

**Isocortex:** frontal pole (FRP), primary motor area (MOp), secondary motor area (MOs), primary somatosensory area (SSp), supplemental somatosensory area (SSs), gustatory area (GU), visceral area (VISC), dorsal auditory area (AUDd), primary auditory area (AUDp), posterior auditory area (AUDpo), ventral auditory area (AUDv), primary visual area (VISp), anterolateral visual area (VISal), anteromedial visual area (VISam), lateral visual area (VISl), posterolateral visual area (VISpl), posteromedial visual area (VISpm), laterointermediate area (VISli), postrhinal area (VISpor), anterior cingulate area, dorsal part (ACAd), anterior cingulate area, ventral part (ACAv), prelimbic area (PL), infralimbic area (ILA), orbital area, lateral part (ORBl), orbital area, medial part (ORBm), orbital area, ventrolateral part (ORBvl), agranular insular area, dorsal part (AId), agranular insular area, posterior part (AIp), agranular insular area, ventral part (AIv), retrosplenial area, lateral agranular part (RSPagl), retrosplenial area, dorsal part (RSPd), retrosplenial area, ventral part (RSPv), posterior parietal association area (PTLp), anterior area (VISa), rostrolateral visual area (VISrl), temporal association area (TEa), perirhinal area (PERI), ectorhinal area (ECT).

**Olfactory areas (OLF):** piriform area (PIR).

**Hippocampal formation (HPF):** hippocampal region (HIP), fields CA1, CA2, CA3, dentate gyrus (DG), entorhinal area, lateral part (ENTl), entorhinal area, medial part (ENTm), parasubiculum (PAR), postsubiculum (POST), presubiculum (PRE), subiculum (SUB), prosubiculum (ProS).

**Cortical subplate (CTXsp):** claustrum (CLA), endopiriform nucleus, dorsal part (EPd), endopiriform nucleus, ventral part (EPv), lateral amygdalar nucleus (LA), basolateral amygdalar nucleus (BLA), basomedial amygdalar nucleus (BMA).

**Cerebral nuclei (CNU):** striatum (STR), caudoputamen (CP), nucleus accumbens (ACB), fundus of striatum (FS), central amygdalar nucleus (CEA), medial amygdalar nucleus (MEA), globus pallidus, external segment (GPe), globus pallidus, internal segment (GPi), bed nuclei of the stria terminalis (BST).

**Thalamus (TH):** ventral anterior-lateral complex (VAL), ventral medial nucleus (VM), ventral posterolateral nucleus (VPL), ventral posterolateral nucleus, parvicellular part (VPLpc), ventral posteromedial nucleus (VPM), ventral posteromedial nucleus, parvicellular part (VPMpc), posterior triangular thalamic nucleus (PoT), medial geniculate complex, dorsal part (MGd), medial geniculate complex, ventral part (MGv), medial geniculate complex, medial part (MGm), lateral geniculate complex, dorsal part (LGd), lateral posterior nucleus (LP), posterior complex (PO), anteromedial nucleus (AM), interanterodorsal nucleus (IAD), lateral dorsal nucleus (LD), mediodorsal nucleus (MD), submedial nucleus (SMT), paraventricular nucleus (PVT), nucleus of reuniens (RE), central medial nucleus (CM), paracentral nucleus (PCN), central lateral nucleus (CL), parafascicular nucleus (PF), reticular nucleus (RT).

**Hypothalamus (HY):** subthalamic nucleus (STN), zona incerta (ZI).

**Midbrain (MB):** substantia nigra, reticular part (SNr), midbrain reticular nucleus (MRN), superior colliculus, motor related (SCm), periaqueductal grey (PAG), anterior pretectal nucleus (APN), red nucleus (RN), pedunculopontine nucleus (PPN), dorsal nucleus raphe (DR).

**Pons:** parabrachial nucleus (PB), pontine grey (PG), pontine reticular nucleus, caudal part (PRNc), tegmental reticular nucleus (TRN), pontine reticular nucleus (PRNr), locus ceruleus (LC).

**Medulla (MY):** cuneate nucleus (CU), gigantocellular reticular nucleus (GRN), inferior olivary complex (IO), intermediate reticular nucleus (IRN), magnocellular reticular nucleus (MARN), parvicellular reticular nucleus (PARN), spinal nucleus of the trigeminal, caudal part (SPVC), spinal nucleus of the trigeminal, interpolar part (SPVI), spinal nucleus of the trigeminal, oral part (SPVO), medullary reticular nucleus, dorsal part (MDRNd).

### Animal care and use

Both male and female transgenic mice from at least postnatal day 56 (P56) were used for all experiments. All animals were housed 3–5 per cage and maintained on a 14 h:10 h light:dark cycle, in a humidity- and temperature-controlled room (humidity at ~40%, temperature at ~21 °C) with water and food available ad libitum. All experimental procedures related to the use of mice were conducted with approved protocols in accordance with NIH guidelines, and were approved by the Institutional Animal Care and Use Committee (IACUC) of the Allen Institute for Brain Science.

### Transgenic mice and sparse labelling

All transgenic crosses are listed in Supplementary Tables 1, 4. Data for systematic characterization of the expression pattern of each transgenic mouse line can be found in the Allen Transgenic Characterization database (http://connectivity.brain-map.org/transgenic/search/basic).

Induction of CreERT2 driver lines was done by administration by oral gavage of tamoxifen (50 mg ml$^{-1}$ in corn oil) at original (0.2 mg g$^{-1}$ body weight) or reduced dose for 1 d in an adult mouse. The dosage for mice aged P7–P15 is 0.04 ml. Mice can be used for experiments at two or more weeks after tamoxifen dosing. We found optimal tamoxifen doses for sparse labelling in each case using serial two photon tomography (STPT)[10,33] to quickly screen for brain-wide transgene expression. The specific dose of tamoxifen to induce sparse labelling in each CreERT2 driver line is shown in Supplementary Table 1.

### fMOST imaging

In summary, a GFP-labelled brain is first embedded in resin. The resin-embedded GFP fluorescence can be recovered through chemical reactivation[47] provided by adding $Na_2CO_3$ in the imaging water bath. Thus, a line-scanning block-face imaging system can be used to maximize imaging speed. Following imaging of the entire block face, the top 1 μm of tissue is sliced off with a diamond knife, exposing the next face of the block for imaging. For the entire mouse brain, a 15–20 TB dataset containing about 10,000 coronal planes of 0.2–0.3 μm $xy$ resolution and 1 μm $z$ sampling rate is generated within 2 weeks.

All tissue preparation has been described previously[48]. Following fixation, each intact brain was rinsed 3 times (6 h for two washes and 12 h for the third wash) at 4 °C in a 0.01 M PBS solution (Sigma-Aldrich). Then the brain was subsequently dehydrated via immersion in a graded series of ethanol mixtures (50%, 70% and 95% (vol/vol) ethanol solutions in distilled water) and the absolute ethanol solution three times for 2 h each at 4 °C. After dehydration, the whole brain was impregnated with Lowicryl HM20 Resin Kits (Electron Microscopy Sciences, cat. no. 14340) by sequential immersions in 50, 75, 100 and 100% embedding medium in ethanol, 2 h each for the first three solutions and 72 h for the final solution. Finally, each whole brain was embedded in a gelatin capsule that had been filled with HM20 and polymerized at 50 °C for 24 h.

The whole brain imaging is realized using a fMOST system. The basic structure of the imaging system is the combination of a line-scanning upright epifluorescence microscope with a mechanical sectioning system. This system runs in a line-scanning block-face mode but updated with the principle of chemical sectioning to obtain better image contrast and speed and thus enables high-throughput imaging of the fluorescent-protein-labelled sample (manuscript in preparation). Each time we do a block-face fluorescence imaging across the whole coronal plane ($xy$ axes), then remove the top layer ($z$ axis) using a diamond knife, and then expose next layer, and image again. The thickness of each layer is 1.0 μm. In each layer imaging, we used a strip-scanning ($x$ axis) model combined with a montage in the $y$ axis to cover the whole coronal plane[49]. The fluorescence, collected using a microscope objective, passes a bandpass filter and is recorded with a TDI-CCD camera. We repeat these procedures across the whole sample volume to obtain the required dataset.

The objective used is a 40× water-immersion lens with numerical aperture 0.8 to provide a designed optical resolution (at 520 nm) of 0.35 μm in the *xy* axes. The imaging gives a sample voxel of 0.35 × 0.35 × 1.0 μm to provide proper resolution to trace the neural process. The voxel size may vary for difference objectives. Other imaging parameters for GFP imaging include an excitation wavelength of 488 nm, and emission filter with passing band 510–550 nm. The fMOST is a two-colour imaging system. The green channel is used to obtain the complete morphology of neurons, and the red channel is used to obtain the cellular architecture information of propidium iodide (PI) staining[20].

## Complete neuronal morphology reconstruction

We used Vaa3D, an open-source, cross-platform visualization and analysis system, for the tasks of reconstructing neuronal morphologies. To efficiently and effectively deal with the mouse whole-brain imaging data, we incorporated several enabling modules into Vaa3D, such as TeraFly, TeraVR and a number of other supporting tools. TeraFly[50] supports visualization and annotation of multidimensional imaging data with virtually unlimited scales. A user can flexibly choose to work at a specific region of interest (ROI) with desired level of detail (LoD). TeraVR[51] is an annotation tool for immersive neuron reconstruction that has been proved to be critical for achieving precision and efficiency in morphology data production. It creates stereo visualization for image volumes and reconstructions and offers an intuitive interface for the user to interact with such data. Both TeraFly and TeraVR are seamlessly integrated in Vaa3D and can be used combinedly and flexibly. From reconstructions (in SWC file format), morphological quantification statistics is obtained to characterize neurons. A quality control process identifies errors based on morphological indicators and does corrections in a feedback setting. The quality control process then refines the skeleton location with Mean-Shift[52] and performs pruning focused on terminal location refinement. When needed, auto-refinement fits the tracing to the centre of fluorescent signals. The whole process ends with SWC resampling and registration. The final reconstruction of each neuron is a valid single tree without breaks, loops, multiple branches from a single point, and so on.

## Registration to CCF

We used mBrainAligner based on BrainAligner[53] to perform 3D registration from fMOST images (subject) to the average mouse brain template of CCFv3 (target) (Extended Data Fig. 4a). The main steps are: (1) fMOST images were first downsampled by 64 × 64 × 16 ($x × y × z$) to roughly match the size of the target brain. (2) The stripe artefacts in fMOST images from diamond-knife cutting and the imaging process were eliminated by using log-space frequency notch filter. (3) The dense outer-contour feature points of target and subject brain (about 1,500 points per brain) were uniformly sampled from the brains' outer contour obtained using adaptive threshold, and then affine-aligned using a reliable landmark points matching algorithm to ensure the subject brain has the same position, orientation and scale as the target brain. (4) Intensity was normalized by matching the local average intensity of subject image to that of target image in a sliding window manner with patch size 41 × 41 × 41 and stride 1. (5) For the target brain, 1,744 landmarks corresponding to the points of high curvature (corners or junction of different brain compartments) in CCFv3 annotation image were detected via 2.5D Harris corner detector. On the basis of a combination of texture, shape context and deep-learning-derived features, mBrainAligner established the correspondence between target and subject brain by iteratively deforming these target landmarks to fit the subject image, and accomplished the local alignment using the smooth-thin-plate-spline (STPS). (6) Finally, to ensure the accuracy of registration, automatic registration results were examined in the semi-automatic registration module of mBrainAligner, and if necessary, the boundaries of the brain region were further optimized in a manual or semi-automatic way. Once images were aligned, the reconstructed neurons and somas were warped to CCF space using the generated deformation fields.

## Processing single-cell morphology data

Pre-processing of SWC files: SWC files were processed and examined with Vaa3D plugins to ensure topological correctness: sorted single tree with root node as soma. Terminal branches < 10 pixels were pruned to remove artifacts. SWC files were resampled with a step size of 64 (*x*), 64 (*y*) and 16 (*z*) before registration.

Quantification of axon projection patterns: to analyse the distribution and amount of axon in brain-wide targets following registration to the CCFv3, we used a manually curated set of 316 non-overlapping structures at a mid-ontology level that are most closely matched in size or division. Ipsilateral and contralateral sides of brain regions were calculated separately.

Morphological features: axonal and dendritic morphological features, defined according to L-measurement[54], were calculated using Vaa3D plugin "global_neuron_feature". Selected features include Axon global: 'Overall Width', 'Overall Height', 'Overall Depth', 'Total Length', 'Euclidean Distance', 'Max Path Distance', 'Number of Branches'; Axon local: 'Total Length', 'Number of Branches'; Dendrite: 'Overall Width', 'Overall Height', 'Overall Depth', 'Total Length', 'Max Euclidean Distance', 'Max Path Distance', 'Number of Branches', 'Max Branch Order'.

Local axons were defined as axon arbors within 200 μm from the somata. Local axons and dendrites were rotated based on principal component analysis (PCA) so dimensions were aligned with the largest to smallest spans. Then shifting was performed to localize somata at the origin of coordinates.

## mBrainAnalyzer

The mBrainAnalyzer toolbox (also named neuro_morpho_toolbox), which was developed for analysis of full neuron morphology, includes multiple modules for feature quantification, arbor detection, statistical analysis and visualization. In addition to morphological features (for example, total length, angle of branches, and so on), this toolbox also quantifies projection intensities at branch length level and number of terminal levels. Using the arbor detection module, one can define sub-cellular components of a neuron as the granularity. Analysis and visualization can be performed at both whole-cell and arbor levels.

## Arbor detection and partition

We detected and partitioned a series of neuronal arbors out of each neuron reconstruction using a graph-partition clustering method. First, as a neuron consists of a number of topologically connected reconstruction nodes, the neuron was viewed as a graph, where every reconstruction node (unit) in the neuron was connected with its parent node with an edge specified by the topological connection of the parent-child pair with the edge weight, or 'similarity' *s*, set to be the exponential of the negative 3D Euclidean distance, *d*, of these two nodes, that is, $s = \exp(-d)$. Then, we considered the normalized graph-cut method[55] to extract 'clusters' of reconstruction nodes so that the within-cluster 'total similarity' of nodes would be maximized and cross-cluster total similarity would be minimized. As a result, each such coherent cluster corresponds to one neuron arbor, which was also visually checked to ensure its correctness. Third, to automatically determine the number of such clusters, for a presumed number of clusters, we calculated the normalized score of total cross-cluster similarity divided by the total within-cluster similarity, followed by trial testing a range (from 2 to 8) of such presumed cluster-numbers to determine the optimal number that would minimizes this normalized score. In the final result, the detected arbor that contains the soma is called soma arbor; the remaining arbors are called non-soma arbors.

## Feature quantification of cortical arbors

We divided the cortex into consecutive 100-μm thick coronal slices. Anchor points were evenly sampled along the outer border of each slice, with normal vectors perpendicular to the local cortical surface

and pointing to the inside of the brain. Nodes of arbors were assigned to their neighbour anchors and projected onto the surface by corresponding normal vectors. Depth of nodes were determined by the length of projection along normal vectors. We also estimated the area of an anchor by their distance to neighbour anchors and slice thickness. The 2D cortical area of an arbor was determined by the total areas of unique anchors occupied by its nodes. To determine the radius of an arbor, we assigned arbor 'centre' as the node that has the shortest average distance to other nodes. Radius was determined by a growing sphere until 70% of segments were inside it. For neurons with tufted apical dendrite, we vertically shifted the arbors, so the top of apical dendrites reached L1. We manually confirmed that all tufted apical dendrites reached L1 in the original image.

## Clustering of cortical axon arbors of core-type thalamic neurons
For local (soma-neighbouring) arbors, the following features were used for clustering: '2d_area', 'total_length', 'radius', 'depth_mean', 'depth_std'. We performed PCA to reduce the effect of noise. Top principal components (PCs) were selected to recover 95% of variance. We applied uniform manifold approximation and projection (UMAP) dimension reduction using the Python package 'UMAP'[56]. The 'n_neighbors' parameter was set at 15. $k$-Means clustering was performed using the UMAP embeddings as input.

## Neuron-beta
We developed the Neuron-beta metric by borrowing the concept of the beta value from the finance field[57]. For each group, defined by brain areas and/or cortical layers, we calculated the average of mesoscale experiments as $M = [m_1, ..., m_p]$, $p$ = number of brain areas. For one single cell $S = [s_1, ..., s_p]$, we define the neuron-beta value as:

$$\text{Neuron beta} = \frac{\text{Cov}(M, S)}{\text{Var}(M)}$$

## Clustering of cortical L5 ET neurons
Projection strength is defined as ln(axon length in mm). Strength values for regions with axon length below 1 mm were set as 0. Only non-cortical areas were included. Dimension reduction was performed by PCA followed by 2D UMAP. Top PCs which explained > 90% variance were used as input of UMAP. Hierarchical clustering was performed using UMAP embeddings. Minimum branch length for clusters was manually determined.

## Clustering of cortical L6 *Car3* and claustral neurons
Data normalization: morphological features were normalized by the mean and standard variation in a feature-wise manner. Projection pattern features were defined as ln(axon length in mm). For regions with axon length below 1 mm, projection pattern feature values are set as 0. Soma locations were flipped to the same hemisphere.

Similarity metrics: for each feature set, we first calculated the Euclidean distance matrix. Then a ranked $k$-nearest neighbour (KNN) matrix was created. We then applied the shared nearest neighbor (SNN) approach to measure the similarity between each pair of samples $x_i$ and $x_j$. The SNN metric was defined as the maximum average rank among their common neighbours:

$$S(x_i, x_j) = \max_{v \in \text{NN}(x_i) \cap \text{NN}(x_j)} \left\{ k - \frac{1}{2} \left[ \text{rank}_{\text{NN}(x_i)}(v) + \frac{1}{2}(\text{rank}_{\text{NN}(x_j)}(v)) \right] \right\}$$

where NN is nearest neighbour and $v$ is a neuron from the dataset. Similarity scores were set as 0 for pairs with non-overlapping KNN sets and a weighted SNN graph was created.

Co-clustering analysis: the co-clustering matrix for each feature set was calculated by iterative random sampling. During each iteration, 95% of samples were randomly selected to create an SNN graph. We then applied the Fast-greedy community detection algorithm using the Python package python-igraph for clustering assignment. For each pair of samples, the co-clustering score was defined as the times of co-clustering normalized by the iterations of co-occurring. Resampling was performed 1,000 times to reach saturation. The overall co-clustering matrix is a weighted average of the four feature sets. Agglomerative clustering was performed on the co-clustering matrix to get clusters.

Outlier removal: outliers were detected by comparing the Euclidean distance between a sample and the other samples with the same cluster identity. We used overall within-cluster distance as the background distribution. Samples with significantly higher (one-sided Mann–Whitney test) within-cluster distance were filtered out as outliers. Agglomerative clustering was performed for the remaining co-clustering matrix. This process iterated until no new outlier could be detected.

Characterization of cell types: for each feature set, we performed two-sided Mann-Whitney tests: claustrum versus cortical neurons; each cluster versus other clusters. $P$-values were adjusted by Bonferroni correction.

## Anterograde tracing and retrograde labelling
For anterograde projection mapping, we injected AAV2/1-pCAG-FLEX-EGFP-WPRE-pA into CLA, SSs or SSp of *Gnb4-IRES2-Cre* or *Gnb4-IRES2-CreERT2* mice at P37–P65. Stereotaxic injection procedures were performed as previously described[10]. For the *Gnb4-IRES2-CreERT2* mice, tamoxifen induction was conducted one week after injection at full dose (0.2 mg per g body weight) for 5 consecutive days. Mice survived 3 weeks (or 4 weeks for the tamoxifen-induced mice) after injection, and brains were perfused and collected for TissueCyte imaging.

For retrograde labelling, we injected several different types of retrograde viral tracers, including AAV2-retro-EF1a-dTomato, AAV2-retro-EF1a-Cre[58], RVdGdL-Cre, RVdL-FlpO[59] or CAV2-Cre[60] into specific target regions of defined transgenic mice (Supplementary Table 4). RFP+ or RFP+/GFP+ cells from defined source regions were collected for scRNA-seq using the procedure described below. Stereotaxic injection procedures were performed as described[10]. Mice were injected at P40 or older, and survived for 16–31 days after injection.

## scRNA-sequencing, clustering and mapping
Cells from transgenic mice or transgenic mice injected with retrograde tracers were collected by microdissection of different cortical regions. Single-cell suspensions were generated and cells were collected using fluorescence activated cell sorting (FACS). FACS gates were selective for cells with fluorescent protein expression from transgenic and/or viral reporters.

Cells were then frozen at −80 °C and were later processed for scRNA-seq using the SMART-Seq v4 method[4]. After sequencing, raw data was quantified using STAR v2.5.3[61] and were aligned to both a Ref-Seq transcriptome index for the mm10 genome, and a custom index consisting of transgene sequences. PCR duplicates were masked and removed using the STAR option bamRemoveDuplicates. Only uniquely aligned reads were used for gene quantification. Gene read counts were quantified using the summarizeOverlaps function from R GenomicAlignments package (RRID: SCR_018096)[62] using both intronic and exonic reads, and quality control was performed as described[4].

Clustering was performed using in-house developed R package scrattch.hicat (available via GitHub: https://github.com/AllenInstitute/scrattch.hicat). The Retro-seq cells where mapped to the most correlated cell types in the Cortex/HPF taxonomy[31] based on a set of 5,981 cell-type-specific markers using the map_sampling function from the scrattch.hicat package. Only the SMART-Seq dataset from the reference taxonomy is used for mapping. All the cells from CLA were mapped to the *Car3* subclass. However, CLA cells were not included as part of the Cortex/HPF taxonomy. To examine more closely the cell-type diversity,

we re-clustered all the original SMART-Seq cells within the *Car3* subclass together with the mapped cells from CLA (Supplementary Table 4; CLA is part of CTXsp), resulting in 8 clusters. The cortical and CLA Retro-seq cells previously mapped to the *Car3* subclass were then re-mapped to the new clusters, using 277 marker genes that discriminate these 8 clusters.

## Reporting summary

Further information on research design is available in the Nature Research Reporting Summary linked to this paper.

## Data availability

The fMOST image datasets (https://download.brainimagelibrary.org/biccn/zeng/luo/fMOST/) of all mouse brains used in this study, as well as the original and CCFv3 registered single neuron reconstructions (https://doi.org/10.35077/g.25), are available at BICCN's Brain Image Library (BIL) at the Pittsburgh Supercomputing Center (www.brainimagelibrary.org). The single-neuron reconstructions, the CCFv3 registered version of these reconstructions, as well as 3D navigation movie gallery of these data are available at SEU-ALLEN Joint Center, Institute for Brain and Intelligence (https://braintell.org/projects/fullmorpho/). Mesoscale AAV-tracing data (including high resolution images, segmentation, registration to CCFv3 and automated quantification of injection size, location, and distribution across brain structures) are available through the Allen Mouse Brain Connectivity Atlas portal (http://connectivity.brain-map.org/). Expression patterns of transgenic mouse lines can be found in the Allen Transgenic Characterization database (http://connectivity.brain-map.org/transgenic/search/basic). Retro-seq SMART-Seq v4 data are available at the NCBI Gene Expression Omnibus (GEO) under accession GSE181363.

## Code availability

Vaa3D (version 3.604), TeraFly (version 2.5.101) and TeraVR (version bundled with Vaa3D) are available at the Vaa3D github, https://github.com/Vaa3D, with both source code and binary executable. The mBrainAligner package is available via https://github.com/Vaa3D/vaa3d_tools/blob/master/hackathon/mBrainAligner. Python package neuro_morpho_toolbox (https://github.com/pengxie-bioinfo/neuro_morpho_toolbox), also known as mBrainAnalyzer, is used for full morphology analysis. Custom data analysis notebooks are available via https://github.com/pengxie-bioinfo/BICCN_full_morphology. STAR v2.5.3[60] and R GenomicAlignments package (RRID: SCR_018096)[61] are used for RNA-sequence alignment. R package scrattch.hicat (https://github.com/AllenInstitute/scrattch.hicat) is used for scRNA-seq analysis, including mapping retro-seq cells to reference taxonomy and re-clustering.

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

**Acknowledgements** We thank the in vivo sciences, molecular biology, histology, and imaging teams at the Allen Institute for their technical support. This work was supported by the Allen Institute for Brain Science and by multiple grant awards from institutes under the National Institutes of Health (NIH), including award number R01EY023173 from The National Eye Institute to H.Z., U01MH105982 from the National Institute of Mental Health and Eunice Kennedy Shriver National Institute of Child Health and Human Development to H.Z., and U19MH114830 from the National Institute of Mental Health to H.Z. and Q. Luo. Generation of the TIGRE-MORF mouse line was partly funded by U01MH106008 and U01MH117079 from the National Institute of Mental Health to X.W.Y. The content is solely the responsibility of the authors and does not necessarily represent the official views of NIH and its subsidiary institutes. This work was also funded by an initiative of Southeast University to support Open Science-oriented international collaboration. Southeast University supported the informatics data management and analysis pipeline at the SEU–Allen Joint Center. The work was also funded by the National Natural Science Foundation of China (NNSFC) grants 61890953 to H.G., 61721092 to Q. Luo, 61871411 to L.Q., 91632201 to W. Xie, the University Synergy Innovation Program of Anhui Province GXXT-2019-008 to L.Q., and by the Tiny Blue Dot Foundation to C.K. The Wenzhou Medical University reconstruction team received funding from the EPFL–Blue Brain Project. The authors thank the founder of the Allen Institute, P. G. Allen, for his vision, encouragement and support.

**Author contributions** H.Z. conceptualized the study. H.P. envisioned and led the development of the computational and data analysis platform. M.B.V., T.L.D., B.T. and X.W.Y. generated the TIGRE-MORF (Ai166) mouse line. Z.J.H. provided Fezf2-CreER, Plxnd1-CreER and Tle4-CreER mouse lines. K.E.H., R.L., T.L.D., B.T. and J.A.H. contributed to the generation and characterization of specific transgenic mouse lines. H.G., A. Li, S. Zeng, X.L., Jing Yuan and Q. Luo conducted fMOST imaging. W. Wakeman, S.J., Y.Y., C.H. and R.D. handled the imaging data. L.Q., Yuanyuan Li, L. Li, W. Wan, T.W., F.X., L.N. and H.P. developed methods for registration of fMOST datasets to CCF. Z. Zhou, S.J., Y.Y., Q. Li, Z.R., Yimin Wang and H.P. developed software and hardware tools for data conversion and morphology reconstruction. Yun Wang, X.K., Yaoyao Li, L. Liu, P.L., C.C., W. Xiong, W. Xu, M.Y., Y.S., L.Y., S. Zhang, S. Zhao, A.F., E.S., J.P., Jia Yuan, M.C., Y.D., P.H., P.W., Yaping Wang, G.H., A. Liu and Z.D. performed manual and semi-automatic morphology reconstruction. L. Liu co-developed the neuron reconstruction pipeline at SEU–ALLEN and contributed to analysis of striatal and thalamic cell types. P.X. performed computational classification of morphological types and other data analysis. Z. Zhou, S.K., H-C.K., L.H., Q.O., F.X., Z. Yun, S. Zhao, Z. Zhao and S.A.S. assisted with morphological analysis. L.D. contributed to single-neuron and population-level projection analysis. Y.Z., D.L. and Yun Wang collaborated with H.P. on developing a quality control method for neuron reconstruction. H.P., W. Xie, Z.G. and H.Z. collaborated in setting up the SEU–ALLEN data-production team. K.E.H., Q.W. and J.A.H. conducted anterograde AAV tracing. T.N.N. performed retrograde tracing. Z. Yao, T.N.N. and B.T. conducted scRNA-seq data generation and analysis. S.M. and S.M.S. provided project management. L.E., M.J.H., B.T., L.N., S.A.S., J.A.H., H.G., Q. Luo, H.P., H.Z. and C.K. provided scientific management. H.Z. wrote the manuscript in consultation with all authors.

**Competing interests** The authors declare no competing interests.

## Additional information

**Correspondence and requests for materials** should be addressed to Hanchuan Peng or Hongkui Zeng.

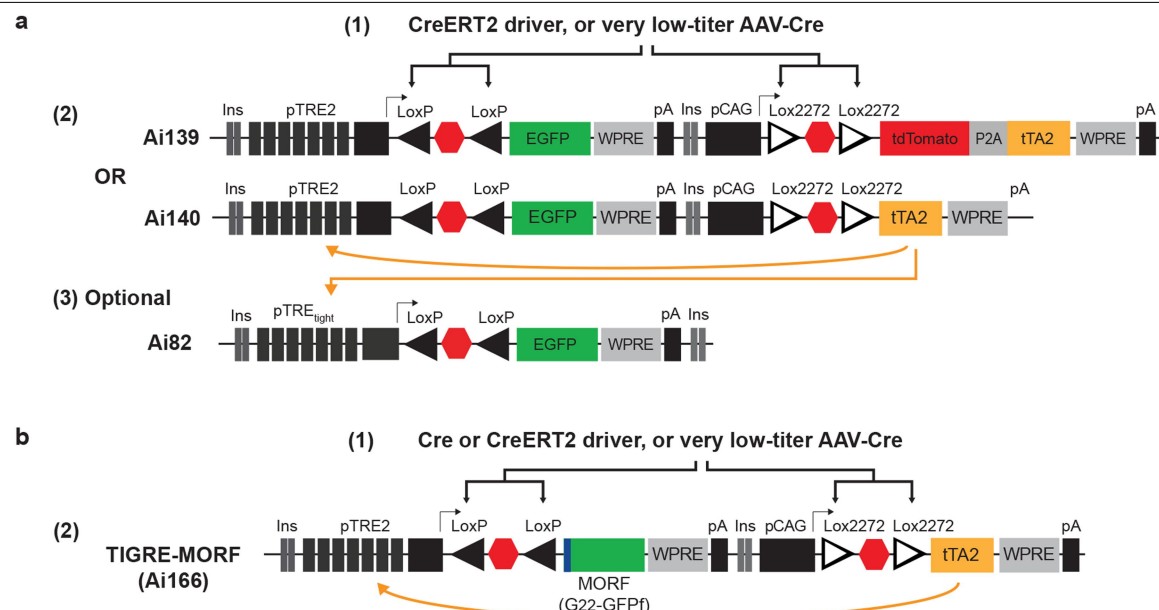

**Extended Data Fig. 1 | Genetic strategy for sparse, robust and consistent brain-wide neuronal labeling. a**, Schematic diagram showing the first approach of sparse and robust labeling, involving the combination of CreERT2 transgenic driver line or Cre-expressing AAV (1) with the GFP-expressing TIGRE2.0 reporter line Ai139 or Ai140 (2). Very low dose tamoxifen induction of CreERT2 (Supplementary Table 1) or very low-titer AAV-Cre delivery results in activation of the reporter in a spatially sparse manner. Transgenic reporter expression of GFP is robust and consistent across different cells. An optional addition is to cross in the GFP-expressing TIGRE1.0 reporter line Ai82 (3), so that the tTA2 from Ai139 or Ai140 will activate the expression of GFP from two alleles – Ai139/Ai140 and Ai82, further increasing the level of GFP within Cre+ cells. **b**, Schematic diagram showing the second approach of sparse and robust labeling, involving the combination of Cre or CreERT2 transgenic driver line or Cre-expressing AAV (1) with the GFP-expressing sparse reporter line TIGRE-MORF (Ai166) (2). In TIGRE-MORF (Ai166), the GFPf transgene is not translated at baseline due to the out-of-frame $G_{22}$ repeat relative to the open reading frame of GFPf, which lacks its own translation start codon. During DNA replication or repair, rare events of stochastic frameshift of the mononucleotide repeat result in correction of the translation frame (*i.e.*, $G_{22}$ to $G_{21}$) and produce expression of the GFPf protein in a small subset of cells. Ai166 exhibits a labeling frequency of 1-5% when crossed to different Cre driver mouse lines[26]. Even with this frequency, we find that combining Ai166 with many Cre driver lines densely expressing the Cre transgene does not produce sufficient

sparsity to readily untangle the axonal ramifications, whereas combining it with Cre lines that are already relatively sparse, or with CreERT2 lines with intermediate dosing level of tamoxifen (Supplementary Table 1), results in very sparse labeling. The use of membrane associated GFPf also enables robust labeling of very thin axon fibers. Leaky background expression of GFP reported in other TIGRE2.0 lines[24] is not present in Ai166 mice due to the strict dependency of translational frameshift for the expression of GFPf reporter, making Ai166 an ideal reporter line for sparse and strong labeling of various neuronal types across the brain. Our labeling strategy using stable and universal transgenic reporter mouse lines coupled with a variety of sparse Cre delivery methods has several advantages. First, the TIGRE2.0-based transgenic reporter lines, especially Ai166 which expresses a farnesylated GFP, produce very bright GFP labeling of axon fibers under fMOST imaging, revealing numerous terminal boutons, an essential requirement for obtaining truly complete morphologies. Second, this strategy enables sparse labeling across multiple regions within the same brain, improving efficiency compared to other methods (*e.g.*, in vivo electroporation or stereotaxic virus injection). Third, the labeling is highly consistent from cell to cell, cell type to cell type, region to region, and brain to brain, reducing variability and enhancing reproducibility. Finally, sparse Cre recombination can be achieved through the use of transgenic Cre or CreERT2 driver lines labeling any neuronal type, or low-dose Cre viral vectors delivered through either local or systemic (*e.g.*, retroorbital) injections.

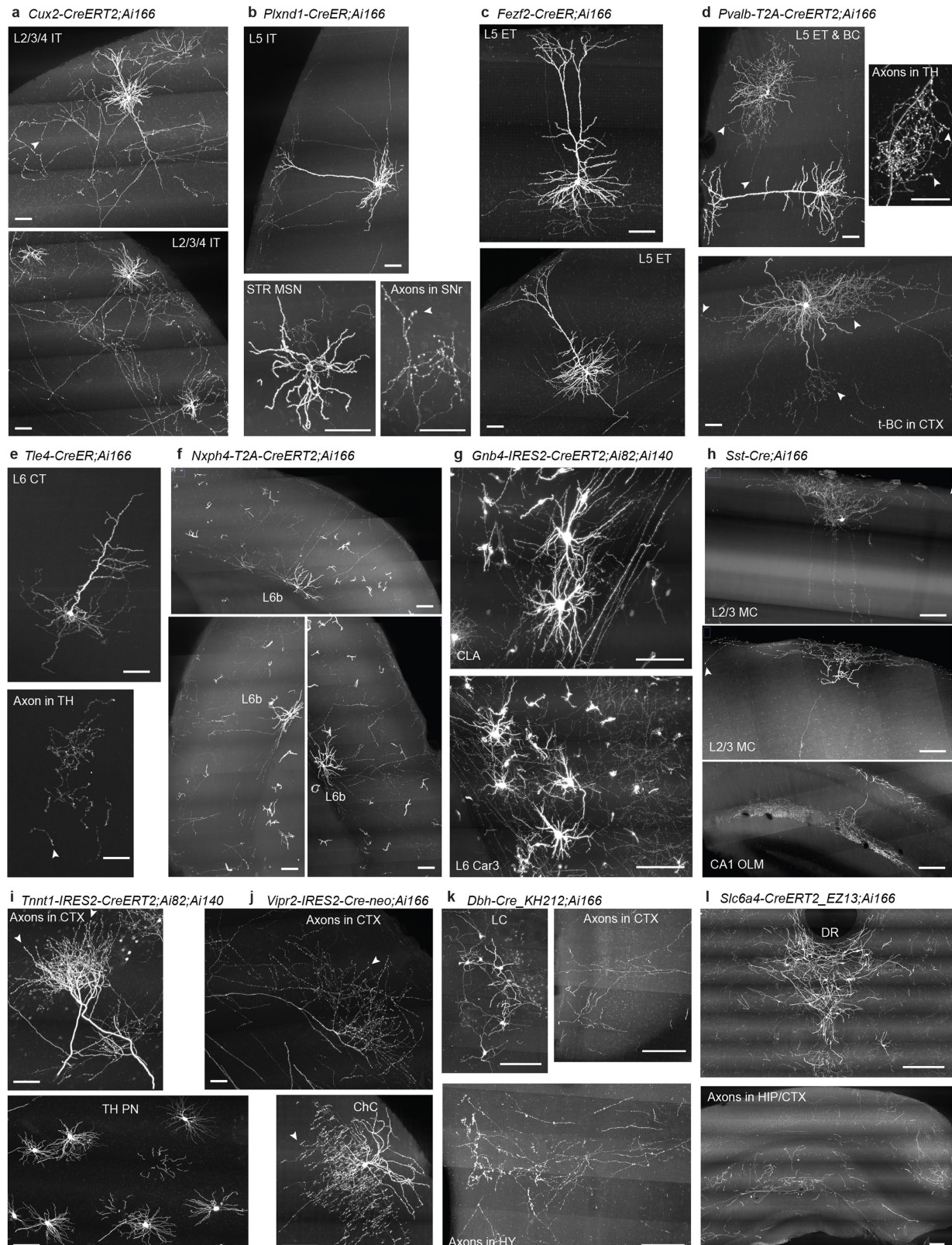

**a** *Cux2-CreERT2;Ai166*

L2/3/4 IT

L2/3/4 IT

**b** *Plxnd1-CreER;Ai166*

L5 IT

STR MSN

Axons in SNr

**c** *Fezf2-CreER;Ai166*

L5 ET

L5 ET

**d** *Pvalb-T2A-CreERT2;Ai166*

L5 ET & BC

Axons in TH

t-BC in CTX

**e** *Tle4-CreER;Ai166*

L6 CT

Axon in TH

**f** *Nxph4-T2A-CreERT2;Ai166*

L6b

L6b

L6b

**g** *Gnb4-IRES2-CreERT2;Ai82;Ai140*

CLA

L6 Car3

**h** *Sst-Cre;Ai166*

L2/3 MC

L2/3 MC

CA1 OLM

**i** *Tnnt1-IRES2-CreERT2;Ai82;Ai140*

Axons in CTX

TH PN

**j** *Vipr2-IRES2-Cre-neo;Ai166*

Axons in CTX

ChC

**k** *Dbh-Cre_KH212;Ai166*

LC

Axons in CTX

Axons in HY

**l** *Slc6a4-CreERT2_EZ13;Ai166*

DR

Axons in HIP/CTX

**Extended Data Fig. 2** | See next page for caption.

**Extended Data Fig. 2 | Sparse, robust and consistent labeling and visualization of the dendritic and axonal arborizations of a wide range of neuronal types.** Images shown are 100-µm maximum intensity projection (MIP) images (*i.e.*, projected from 100 consecutive 1-µm image planes). Arrowheads indicate observed terminal boutons at the end of the axon segments. Number of fMOST imaged brains per mouse line and tamoxifen induction conditions are shown in Supplementary Table 1. Scale bars, 100 µm. **a**, Cortical L2/3/4 IT neurons and their extensive local axon collaterals clearly labeled in a *Cux2-CreERT2;Ai166* brain. **b**, Cortical L5 IT neurons and their local axon collaterals seen in a *Plxnd1-CreER;Ai166* brain. Striatal medium spiny neurons (STR MSN) are also sparsely labeled, and their individual axons are clearly seen in substantia nigra, reticular part (SNr). **c**, Cortical L5 ET neurons and their sparse local axon collaterals seen in a *Fezf2-CreER;Ai166* brain. **d**, Cortical inhibitory basket cells (BC) and translaminar basket cells (t-BC), as well as L5 ET excitatory neurons, seen in a *Pvalb-T2A-CreERT2;Ai166* brain. The L5 ET neurons form driving-type axon clusters with large boutons in the thalamus (TH). **e**, Cortical L6 CT neurons and their characteristic apical dendrites not reaching L1, as well as local axon collaterals and long-range axon projections into thalamus (TH), labeled in a *Tle4-CreER;Ai166* brain. **f**, Cortical L6b neurons and their local axon projections up into L1 seen in a *Nxph4-T2A-CreERT2;Ai166* brain. **g**, *Gnb4+* claustral (CLA) and cortical (L6 *Car3*) neurons with their widely dispersed axon fibers seen in a *Gnb4-IRES2-CreERT2;Ai140;Ai82* brain. **h**, Cortical inhibitory Martinotti cells (MC) and hippocampal CA1 OLM cells labeled in a *Sst-Cre;Ai166* brain. **i**, Thalamic projection neurons (TH PN) with their dense axon terminal clusters in cortex seen in a *Tnnt1-IRES2-CreERT2;Ai82;Ai140* brain. **j**, In a *Vipr2-IRES2-Cre-neo;Ai166* brain, axon clusters from projection neurons in visual thalamic nuclei are seen in cortex (CTX), and a cortical chandelier cell (ChC) is fully labeled with its characteristic axonal branches. *Vipr2-IRES2-Cre-neo;Ai166* also labels axons consistent with projections from retinal ganglion cells[63], which are not shown here. **k**, Noradrenergic neurons labeled in the locus ceruleus (LC), and their long-range axon fibers seen in CTX and hypothalamus (HY) in a *Dbh-Cre_KH212;Ai166* brain. **l**, Serotonergic neurons labeled in the dorsal raphe (DR), and their long-range axon fibers seen in hippocampus (HIP) and CTX in a *Slc6a4-CreERT2_EZ13;Ai166* brain. Overall, it is apparent that these neurons display a remarkable array of dendritic and axonal morphologies. Specifically, in these sparsely labeled brains, cortical IT, ET and CT neurons not only have primary long-range projections but also local axonal branches that are well segregated and clearly identifiable, enabling truly complete reconstruction of the entire local and long-range, cortical and subcortical axonal arborization (**a**–**e**). L5 ET neurons form the 'driving' type of synapses in the thalamus[34,64], which have enlarged and intensely fluorescent boutons (**d**). L6b subplate neurons extend their local axon collaterals upwards into layer 1 (**f**). The axons of thalamic projection neurons form either dense or dispersed clusters in the cortex (**i**, **j**). On the other hand, claustral, noradrenergic and serotonergic neurons have widely dispersed, thin axons that are nonetheless well labeled (**k**, **l**). One can also clearly see individual axons in the substantia nigra from striatal medium spiny neurons (**b**), as well as dense and fine local axonal branches of a variety of cortical interneurons (*e.g.*, basket cells, Martinotti cells and chandelier cells) (**d**, **h**, **j**). Of note, sparsely labeled neurons were frequently observed in other regions of the brain for all of these crosses but are not described in detail here. Each of these brains contains ~100-1,000 labeled neurons (Supplementary Table 1). Thus, tens of thousands of neurons could be reconstructed from these and newly generated datasets in the coming years. The whole brain image datasets are publicly available as a unique resource for the community.

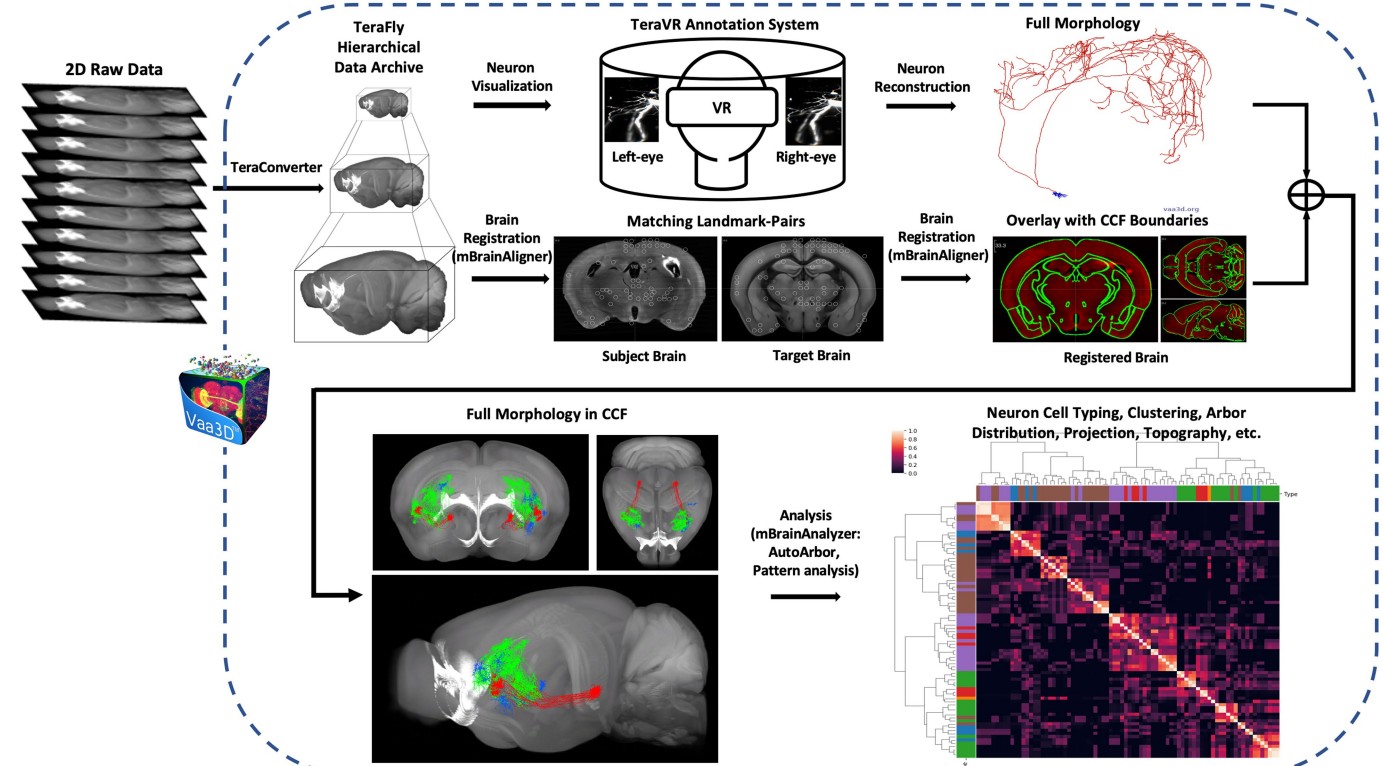

**Extended Data Fig. 3 | Platform and workflow of the brain-wide complete morphology imaging, reconstruction, registration and analysis pipeline.** Each fMOST dataset is first converted to a multi-level navigable TeraFly dataset using TeraConverter, the data formatting tool in the Vaa3D-TeraFly program[50], which allows smooth handling of terabyte-scale datasets. Neuron visualization and reconstruction is then carried out on the TeraFly files. A series of tools, especially those based on the "Virtual Finger" method[65], were developed within Vaa3D to facilitate semi-automated and manual reconstruction. Further, a virtual reality (VR) environment created within Vaa3D, TeraVR, significantly enhances a user's ability to see the 3D relationships among intertwined axonal segments, improving precision and efficiency of reconstruction[51]. Annotators work in the TeraVR annotation system to reconstruct the full morphology of each neuron. After quality control (QC) and manual correction, Vaa3D's deformable model is used to automatically fit the tracing to the center of fluorescent signals. The final reconstructed morphology is completed as a single tree without breaks, loops, or trifurcations. All these data processing,

reconstruction, and workflow control processes are managed using a specific software system for massive scale data production. In parallel, each fMOST dataset is registered to CCFv3 using mBrainAligner, which uses both CLM (Coherent-Landmark-Matching) and LQW (Little-Quick-Warp) modules in brain alignment. Following registration of the whole-brain image dataset to CCFv3, all the reconstructed morphologies from the same brain are also registered for subsequent visualization and quantitative analysis. Registration to CCFv3 enables digital anatomical delineation and spatial quantification of each reconstructed morphology and its compartments (*e.g.*, soma, dendrites, axon arbors). Since neurons are reconstructed from different brains, co-registration to the CCFv3 allows them to be compared and analyzed using a unified framework, mBrainAnalyzer, which automatically detects the arbors of each neuron followed by mapping of these dendritic and axonal arbors onto the standardized CCFv3 space. Morphological features such as length, depth, area, etc., at the whole neuron level are also computed for each arbor-domain for analysis.

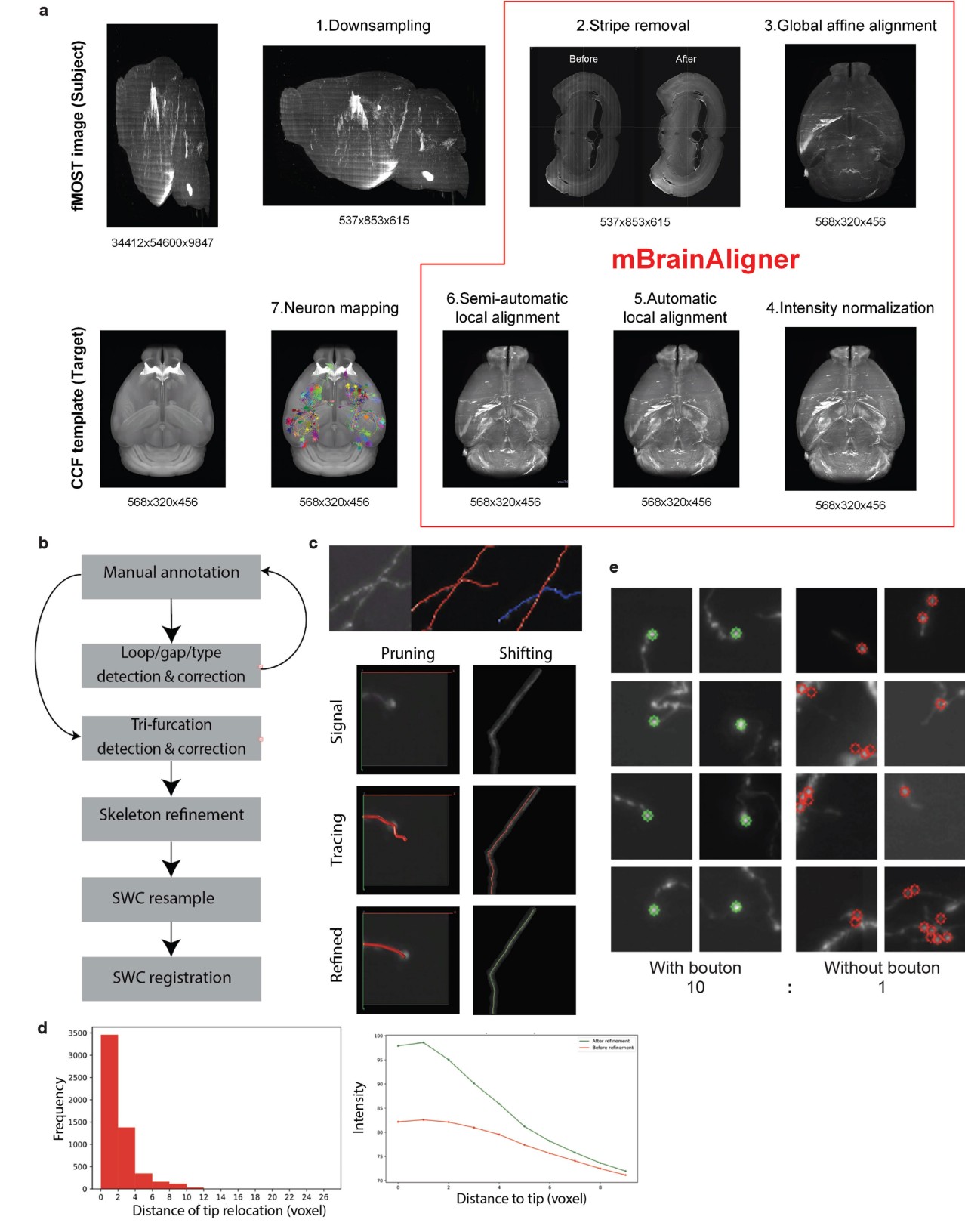

**Extended Data Fig. 4** | See next page for caption.

**Extended Data Fig. 4 | CCF registration and QC of reconstructed morphologies. a**, CCF registration workflow. Pipeline of 3D registration from fMOST image (subject) to average mouse brain template of CCFv3 (target). Numbers below each panel indicate the pixel sizes in the order of X*Y*Z. See Methods for explanation of each step. **b**, Workflow of post-processing process for QC of reconstruction SWC files: 1. automatic detection and correction of basic reconstruction errors including loops, gaps and incorrect node types. 2. Corrections are sent back for manual verification. 3. Automatic detection and correction of trifurcations, which are usually overlapping neurites, instead of branching points. 4. Refinement of SWC files, including pruning of over-traced terminals and shifting skeleton to fit the center of image signals. 5. Resampling of SWC to achieve evenly distributed nodes. 6. SWC registration to the standard CCFv3 mouse brain template. **c**, Examples of trifurcation before (middle) and after (right) correction (blue and red branches do not cross), and examples of refinement before and after pruning (lower left panels) and shifting (lower right panels).

**d**, Refinement leads to more precisely defined axon termination. Upper, distribution of terminal relocation distance by pruning. Lower, radius-decay curve of terminal signals shows that after refinement the axon ends at a brighter spot (indicating a bouton) rather than tapering off. **e**, Examples of axon terminals that end with or without a bouton. We established a stringent QC process that includes ensuring the completeness of reconstructed morphologies. A conventional way to assess the completeness of axon labeling and reconstruction is whether an axon ends at a bouton, as indicated by an enlargement with more intense signal (see arrowheads in Extended Data Fig. 2), or gradually tapers off, the former suggesting a complete labeling[23]. We implemented this assessment in our reconstruction refinement process to identify potential inaccuracies. In our final QC-passed reconstructions we found that the ratio between terminal axon branches with and without a terminal bouton was about 10:1, indicating a high degree of completeness of our reconstructed morphologies.

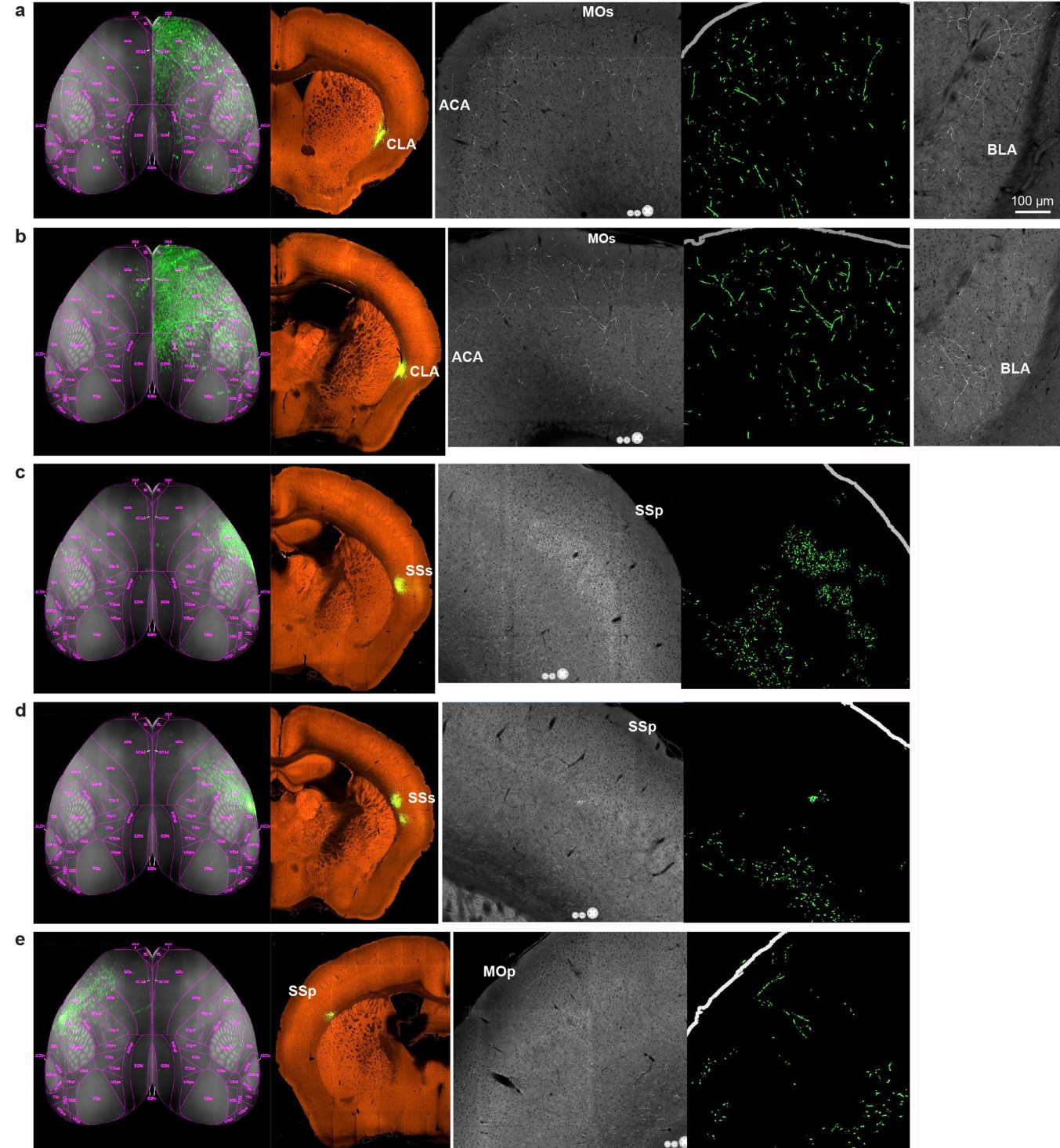

**Extended Data Fig. 5 | Anterograde bulk AAV tracing of projections from _Gnb4+_ neurons in claustrum or lateral cortex. a**–**e**, AAV2/1-pCAG-FLEX-GFP tracer was injected into the claustrum (**a**, **b**), SSs (**c**, **d**) or SSp (**e**) in _Gnb4-IRES2-Cre_ or _Gnb4-IRES2-CreERT2_ mice. Brains were imaged by the TissueCyte STPT system. First panel in each row: top-down view of segmented GFP-labeled axon projections in the cortex. Second panel: injection site. Third panel: the fine axon fibers in a target cortical area. Fourth panel: the segmented image of the third panel to visualize and quantify the axon fibers. Fifth panel in **a** and **b**: axon fibers observed in basolateral amygdalar nucleus (BLA). Full STPT image datasets are available at the Allen Mouse Brain Connectivity Atlas web portal (http://connectivity.brain-map.org/) with the following experiment IDs: **a**, 514505957; **b**, 485902743; **c**, 553446684; **d**, 581327676; **e**, 656688345. These 5 selected datasets were replicates of each other and all had small, spatially specific, injection sites that were located very close to each other. These small bulk injections demonstrate very distinct projection patterns between claustral and cortical _Gnb4+_ neurons.

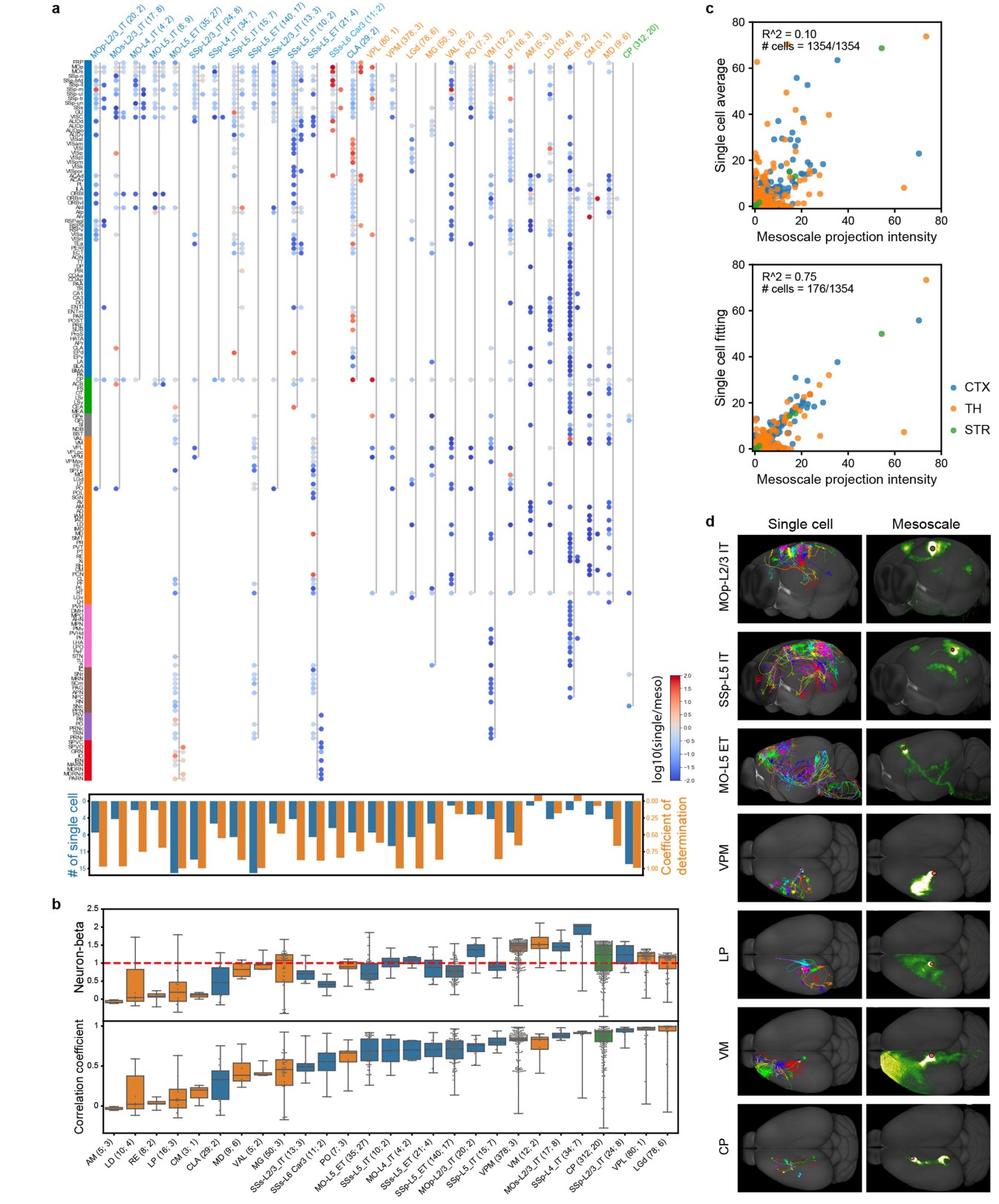

**Extended Data Fig. 6** | See next page for caption.

**Extended Data Fig. 6 | Combination of single neuron morphologies recapitulates population-level mesoscale projection patterns.**
**a**, Comparative projection map of single cells and mesoscale experiments. Individual samples are grouped by brain areas and/or cortical layers based on soma locations (single cell) and injection sites (mesoscale). This dataset covers 14 cortical areas and layers combined, 13 thalamic nuclei and one striatal structure (CP). Each group is represented by a stretch of connected dots with ipsilateral and contralateral targets in the two hemispheres, respectively. Projection intensities in each target region are quantified as ln(NPV × 100 + 1) for mesoscale experiments, where NPV denotes normalized projection volume (Supplementary Table 3), and axon lengths within the target region for single cells. Target regions are defined using thresholds of ln(NPV × 100 + 1) > 0.2 for mesoscale experiments and axon length > 1 mm for single cells. Only target regions present in at least 50% mesoscale experiments or 10% single cells are shown here. Dot colors are scaled by the log10 of single-cell and mesoscale strength ratio. Lower panel, coefficients of determination (orange bars) and number of cells (blue bars) of mesoscale regression by single cells (described in **c**). **b**, Box plots of neuron-beta and correlation coefficients between single cells and group-average of mesoscale data. Individual comparisons shown as swarm plots overlapped with boxes. Box plot specifications: box bounds = 25th and 75th percentile, center = 50th percentile, minima/maxima = center ± 1.5 × IQR (75th percentile – 25th percentile), no whiskers shown. The first and second numbers in the group labels in **a** and **b** indicate the numbers of single cells and mesoscale experiments, respectively. To quantitatively compare the single cell and mesoscale tracer experiments, we calculated the correlation coefficient of each single cell's brain-wide projection weights with the average projection weights from the location-matched mesoscale experiments. The correlation coefficients range from −0.04 (*i.e.*, AM) to 1.00 (*i.e.*, LGd), with a median of 0.69. High correlation coefficients may indicate simple compositions of projecting patterns, *e.g.*, LGd with almost pure VISp projections and CP projecting mainly to either GPe or SNr. Low correlation coefficients may indicate complex composition of projecting patterns, *e.g.*, AM, RE and CM for reasons mentioned below. To compare single cell projection strength relative to mesoscale data, we developed a 'Neuron-beta' metric, as the covariance of a single cell and the average of mesoscale samples, relative to the mesoscale variance. Single cells with Neuron-beta values > 1.5 correlate well with mesoscale data but fluctuate more variably. For example, individual VM

neurons are highly diverse but positively correlated with mesoscale data. Small (< 0.5) positive Neuron-beta values result from low correlation of single cell and mesoscale data. For cell types with Neuron-beta values around 1, single cell and mesoscale data appear to be comparable. **c**, Approximation of mesoscale projections by single cell projection strengths (1,354 cells used) by group-average (upper) or by linear regression (lower). To study how well the mesoscale projection pattern could be broken down to our set of single cells, we performed linear regression with Lasso regularization to reduce the number of single cells with non-zero weights (representative cells). This approach selects a minimal set of single cells and uses weighted summation of single cell axon length to approximate the cell type specific mesoscale axonal weights. The overall coefficient of determination ($R^2$) is 0.75, indicating that mesoscale connectivity is recapitulated well except for the above-mentioned thalamic nuclei. Only 176 out of 1,354 single cells contribute to the regression. These cells represent a minimal set of stereotypes to make up the population level connectivity (see **d**). Averaging across all single cells shows a low level of approximation ($R^2 = 0.10$), suggesting highly diverse morphologies and projection patterns among the single cells. **d**, Visualization of projection patterns constituted by representative cells and mesoscale projection intensities. Overall, the combined single cell projection pattern from a given region (and cortical layer) is highly concordant with that of the mesoscale experiments. There are a few exceptions to this general trend. The combined patterns from single cortical and CLA *Car3* neurons collectively project to more targets than mesoscale experiments, likely due to the richer sampling of single neurons across multiple cortical areas and along the entire extent of CLA than the few mesoscale experiments covered. On the other hand, for several thalamic nuclei (*e.g.*, VAL, VM, AM, RE and CM), single neurons collectively have not captured the full projection patterns from mesoscale experiments. This difference could be due to several reasons: (1) since some of these nuclei are small, the mesoscale experiments may include projections labeled from neighboring nuclei so the single cell data may more accurately represent the true output pattern; (2) the number of reconstructed single neurons is still small and may not fully represent all projection types in a given nucleus; (3) the reconstructed neurons may represent only a subset of the cell types located in these nuclei, and there may be other types of projection neurons not labeled in the Cre lines used here.

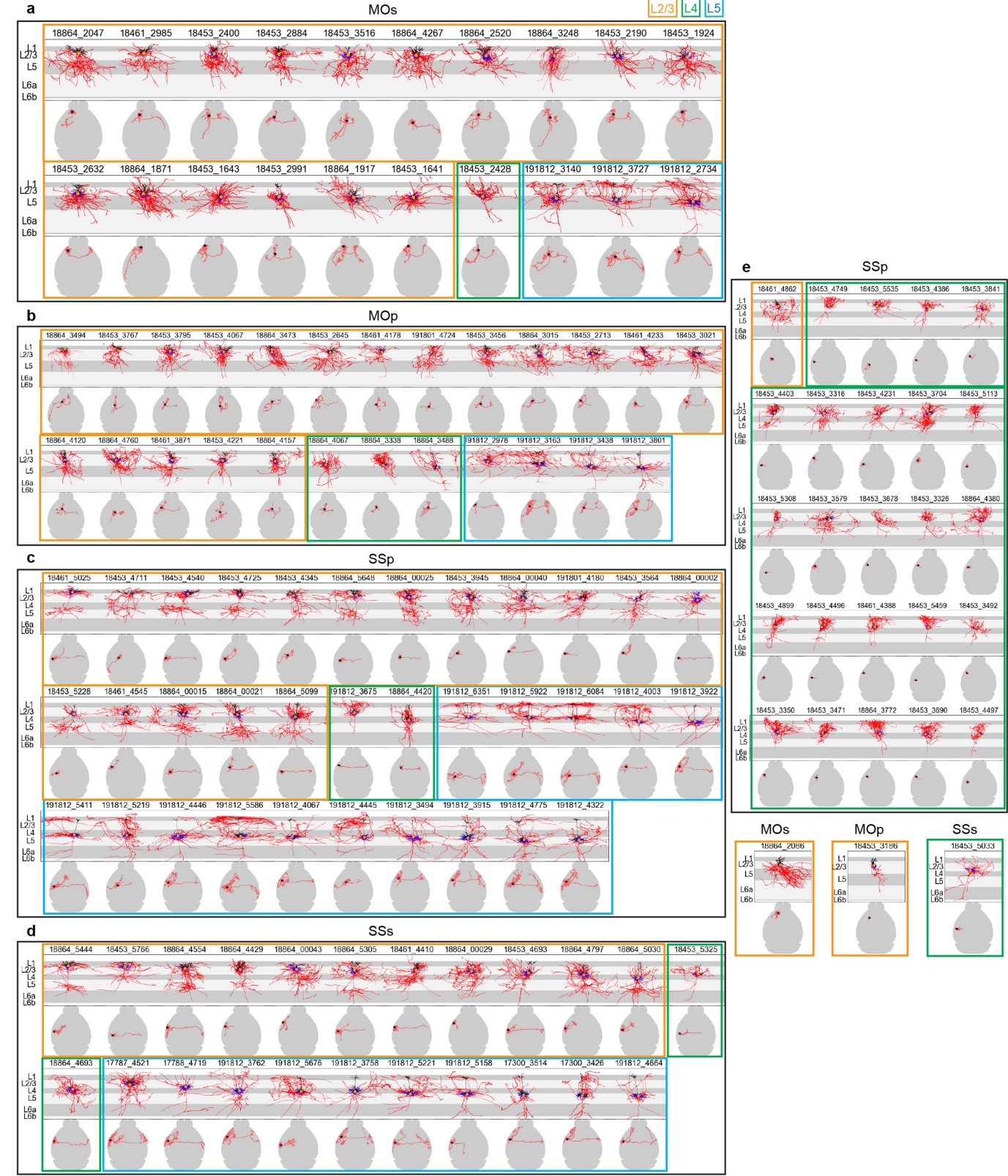

**Extended Data Fig. 7** | See next page for caption.

**Extended Data Fig. 7 | Local morphologies and long-range intracortical projections of cortical L2/3, L4 and L5 IT neurons. a–d,** Comparison of local morphologies (upper panels; apical dendrite in black, basal dendrite in blue, axon in red, soma as an orange dot) and intracortical projections (lower panels; axon in red, soma as a star) for MOs (**a**), MOp (**b**), SSp (**c**) and SSs (**d**) neurons. L2/3, L4 and L5 IT neurons are marked by orange, green and blue boxes, respectively. The L4-like neurons from MOp and MOs are located between L2/3 and L5 since L4 is not delineated in MOp or MOs in CCFv3. Neurons are ordered based on the depths from pial surface of their somas. Gray shadings mark generic layers; however, it should be noted that due to variation in layer thickness in different parts of the cortical areas, the generic layer marking does not necessarily correlate with each neuron's precise soma location. The layer assignment of each neuron's soma location was confirmed by visual inspection of each case. **e,** Reconstructed neurons without long-range axon projections outside of their soma areas. Vast majority of these neurons are SSp L4 IT. Overall, recent studies by scRNA-seq[31], MERFISH[66] and Patch-seq[7] showed that transcriptomically defined cortical IT neuron types are organized by layer, but also exhibit a continuous spatial transition along the cortical depth. Here we arrange the L2/3, L4 and L5 IT neurons according to the depth of their soma from the pial surface, and find that within each region, across depths individual neurons exhibit highly variable long-range projection patterns. We identified 26 cells from SSp and 3 cells from SSs to be in L4. L4 cells have either no apical dendrites (*i.e.*, spiny stellate cells) or a simple apical dendrite that does not branch in L1 (*i.e.*, untufted or star pyramid cells), in contrast to the pyramidal L2/3 cells which have tufted or wide-branching apical dendrites in L1[67]. L2/3 cells have local axons branching in L2/3 and downward into L5, whereas L4 cells have local axons mainly projecting up to L2/3[68]. We also found 4 cells from MOp and MOs with these L4-like features – minimal apical dendrites and upward-projecting local axons, suggesting that these are the L4-like cells located in motor cortex[69] that can also be identified transcriptomically[70]. Consistent with prior notion[67], all but two SSp L4 cells have only local axons but no long-range projections. However, nearly all L4 cells in SSs, MOp and MOs do have axon projections outside of their local area, as we reported before[33].

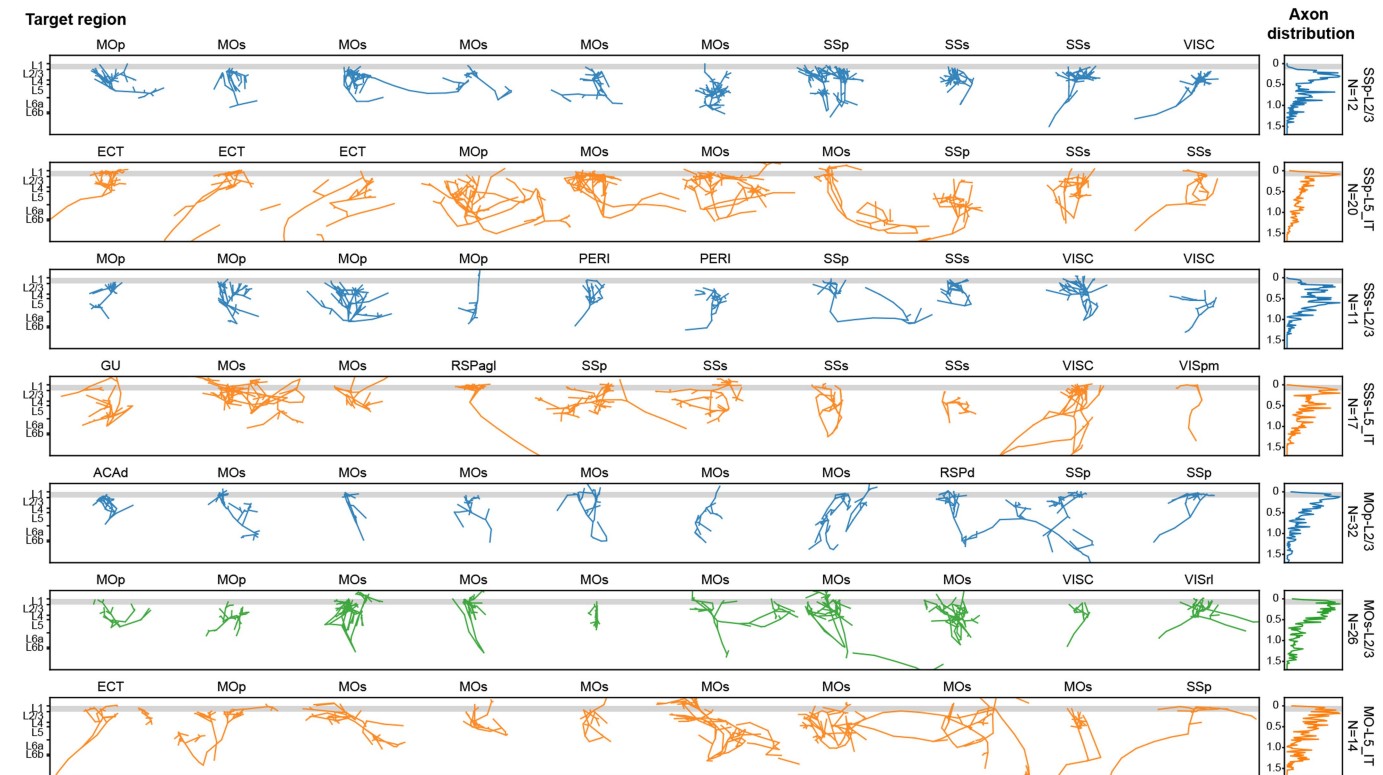

**Extended Data Fig. 8 | Comparison of terminal axon arbor patterns in target cortical regions between L2/3 and L5 IT neurons.** Axon terminals in specific target regions (labeled on top) of L2/3 or L5 IT neurons from SSp (first two rows), SSs (next two rows), MOp and MOs (last three rows). Because not all neurons project to all target regions, axon terminals from any neurons for each target region are combined. For each neuron type from each region, 10 representative axon terminals are shown here, whereas the overall vertical profile of axon distributions (the rightmost panel) is quantified from all axonal terminals (N indicated below each neuron type label). These axon distribution vertical profiles are also presented in Fig. 2d. Due to variation in layer thickness in different parts of the cortical areas, only generic L1 is shaded.

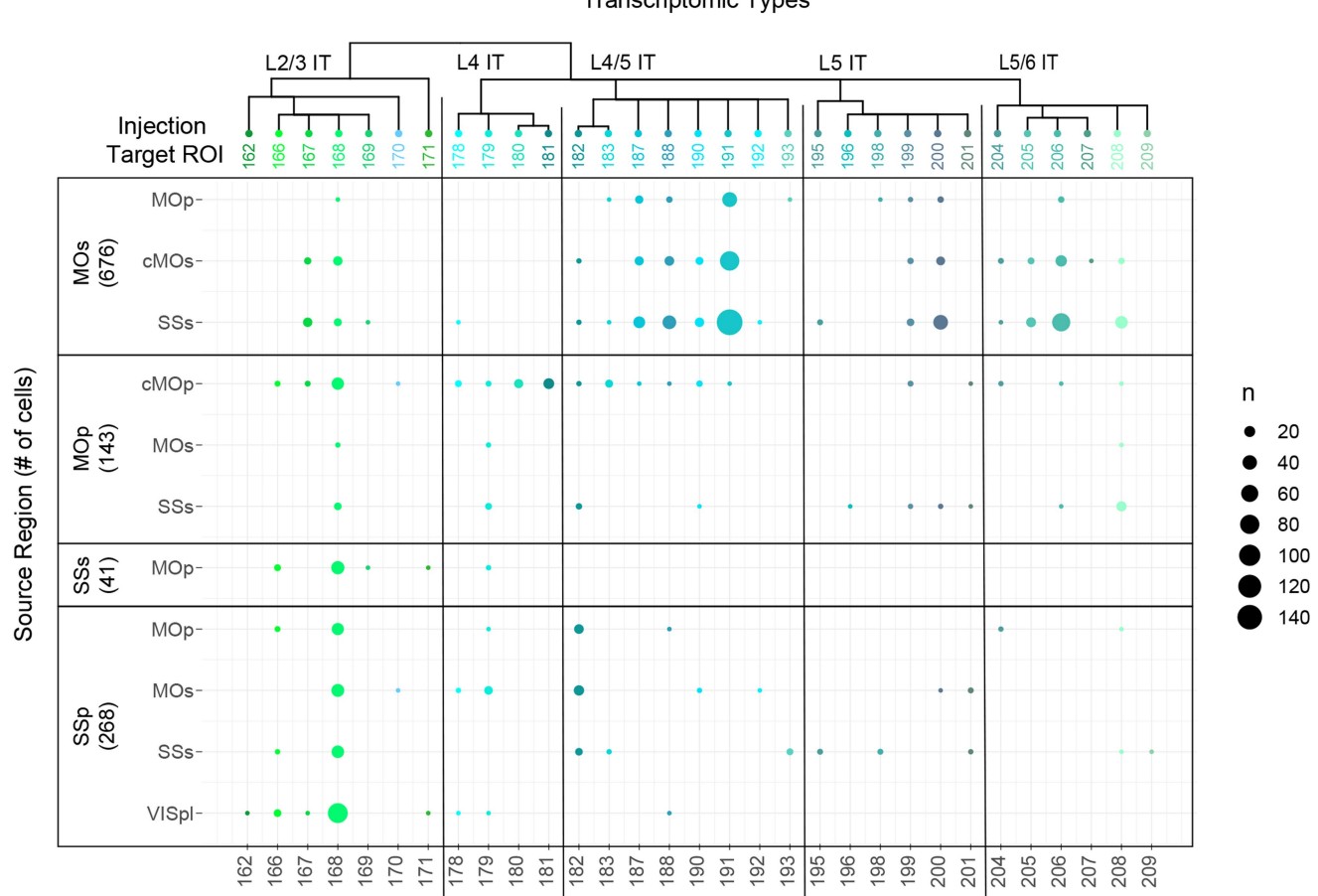

**Extended Data Fig. 9 | Retro-seq characterization of cortical IT neurons across all layers from MOs, MOp, SSp and SSs.** Transcriptomes of retrogradely labeled neurons were obtained by single cell or nucleus RNA-sequencing and then mapped to our transcriptomic taxonomy[31] to identify the transcriptomic type (shown as clusters at the top and bottom of the dot plot) of each neuron. Cells are grouped by their source region. Within each source region, cells labeled from different projection targets (Injection Target region of interest, ROI) are compared, and found to be assigned to a similar subset of transcriptomic types without major distinction between ROIs. cMOs or cMOp denotes contralateral MOs or MOp, respectively.

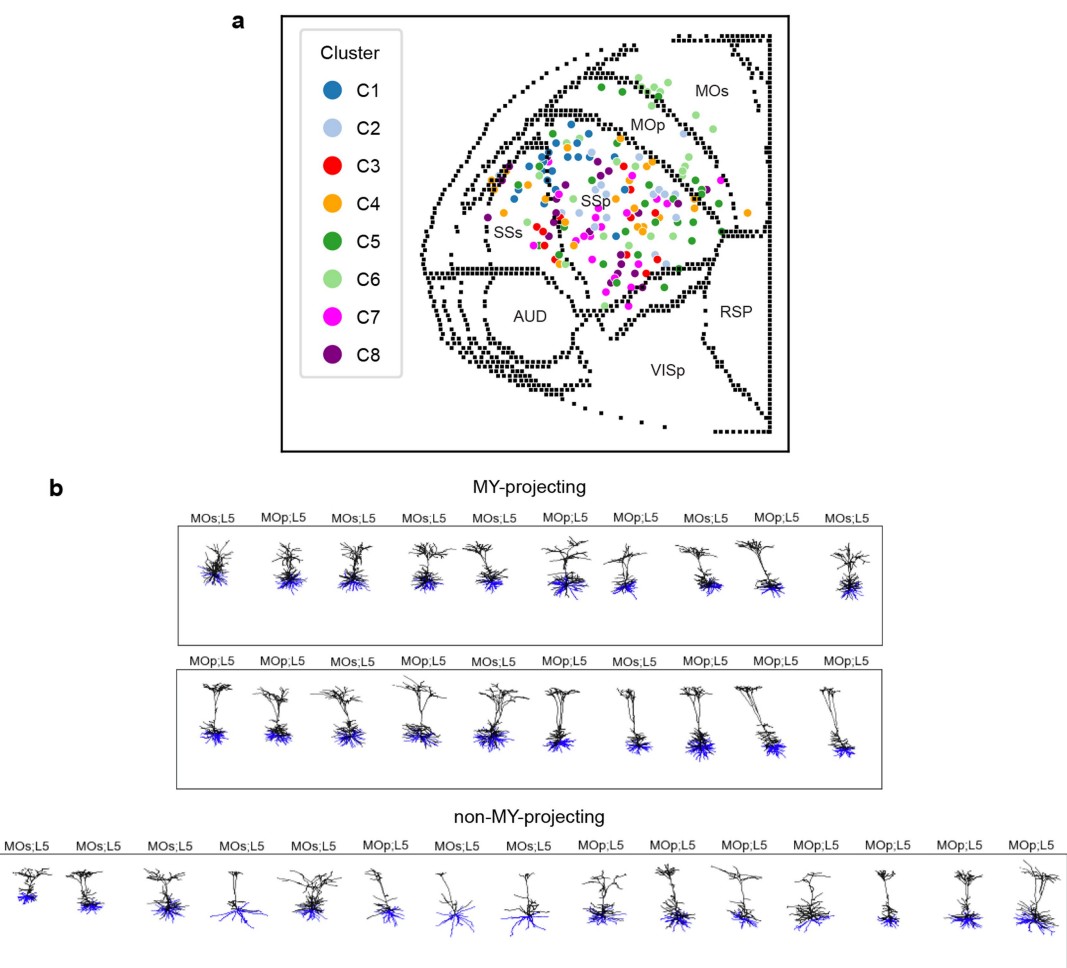

**Extended Data Fig. 10 | Local morphology and long-range projection analysis of cortical L5 ET neurons. a**, Cortical flatmap showing the distribution of cells belonging to each cluster. **b**, Dendritic morphologies of motor cortex (MOp and MOs) L5 ET neurons separated into medulla (MY)-projecting and non-MY-projecting groups. Apical dendrite in black, basal dendrite in blue.

Comparing the dendritic morphologies of MOp and MOs neurons with or without MY projection shows that MY-projecting neurons tend to have denser basal dendrites as well as more extensive and complex apical dendrites that have their first bifurcation points closer to the somas.

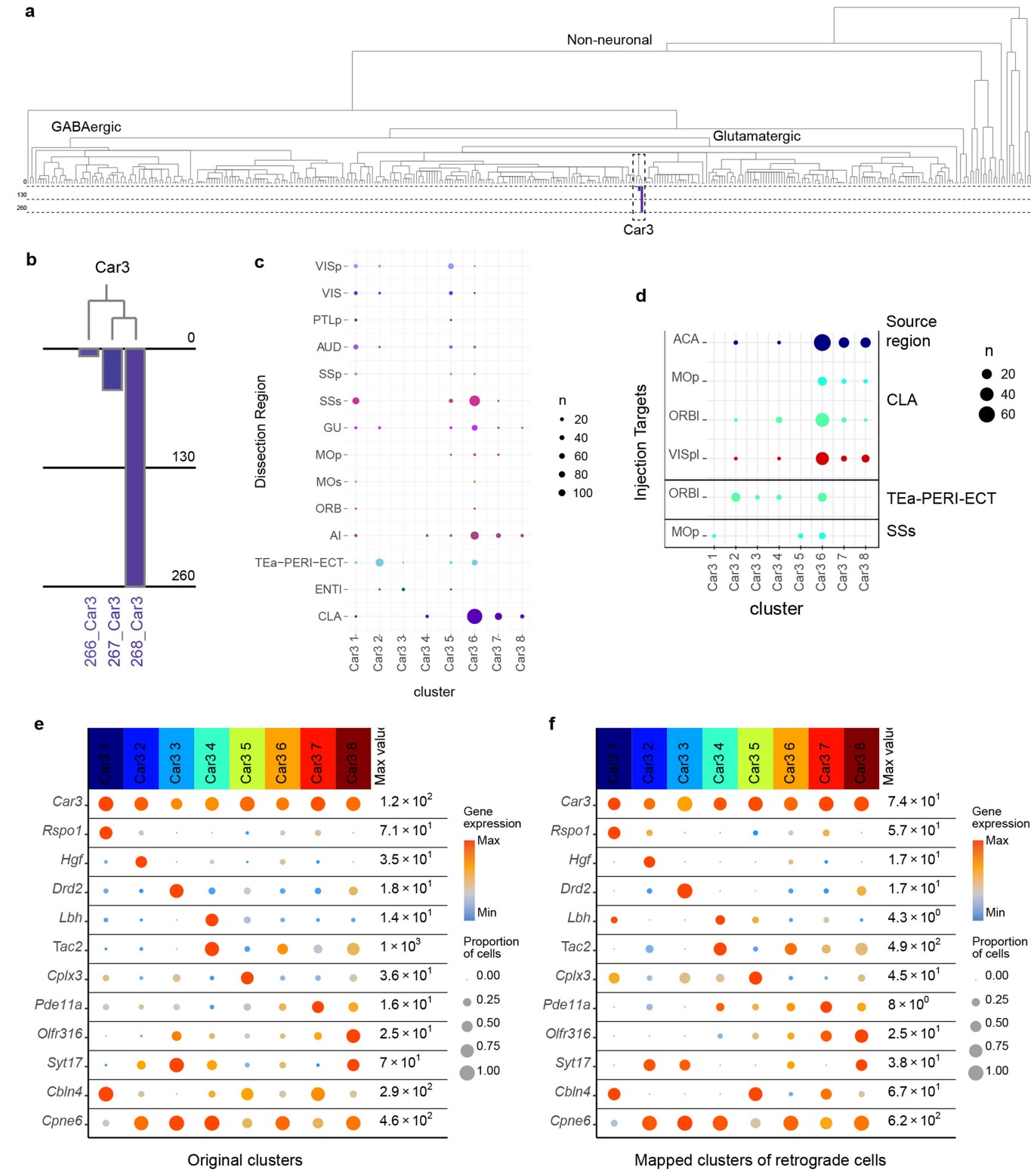

**Extended Data Fig. 11 | Single-cell RNA-seq characterization of *Car3* subclass of cortical and claustral neurons. a**, Transcriptomic taxonomy of the entire mouse isocortex and hippocampal formation[31] reveals a distinct branch of *Car3* subclass (dashed box). Bar graph shows the distribution of Retro-seq cells within the *Car3* subclass **b**, Enlarged view of the taxonomy part within the dashed box in **a** shows the distribution of Retro-seq cells in the 3 clusters of the *Car3* subclass. **c**, Neurons from claustrum (CLA) are also entirely mapped to the *Car3* subclass. We combined all CLA and cortical *Car3* neuron

SMART-Seq transcriptomes and re-clustered them to see if more refined cluster segregation could be obtained, resulting in 8 clusters. Dot plot shows the number of cells from each cortical region or CLA contributing to each cluster. **d**, Remapping of Retro-seq cells from CLA and several cortical regions (TEa-PERI-ECT and SSs) to the 8 new transcriptomic clusters. **e**, **f**, Marker gene expression for different clusters is similar between (**e**) cells contributing to the clustering shown in c and (**f**) Retro-seq cells shown in **d**.

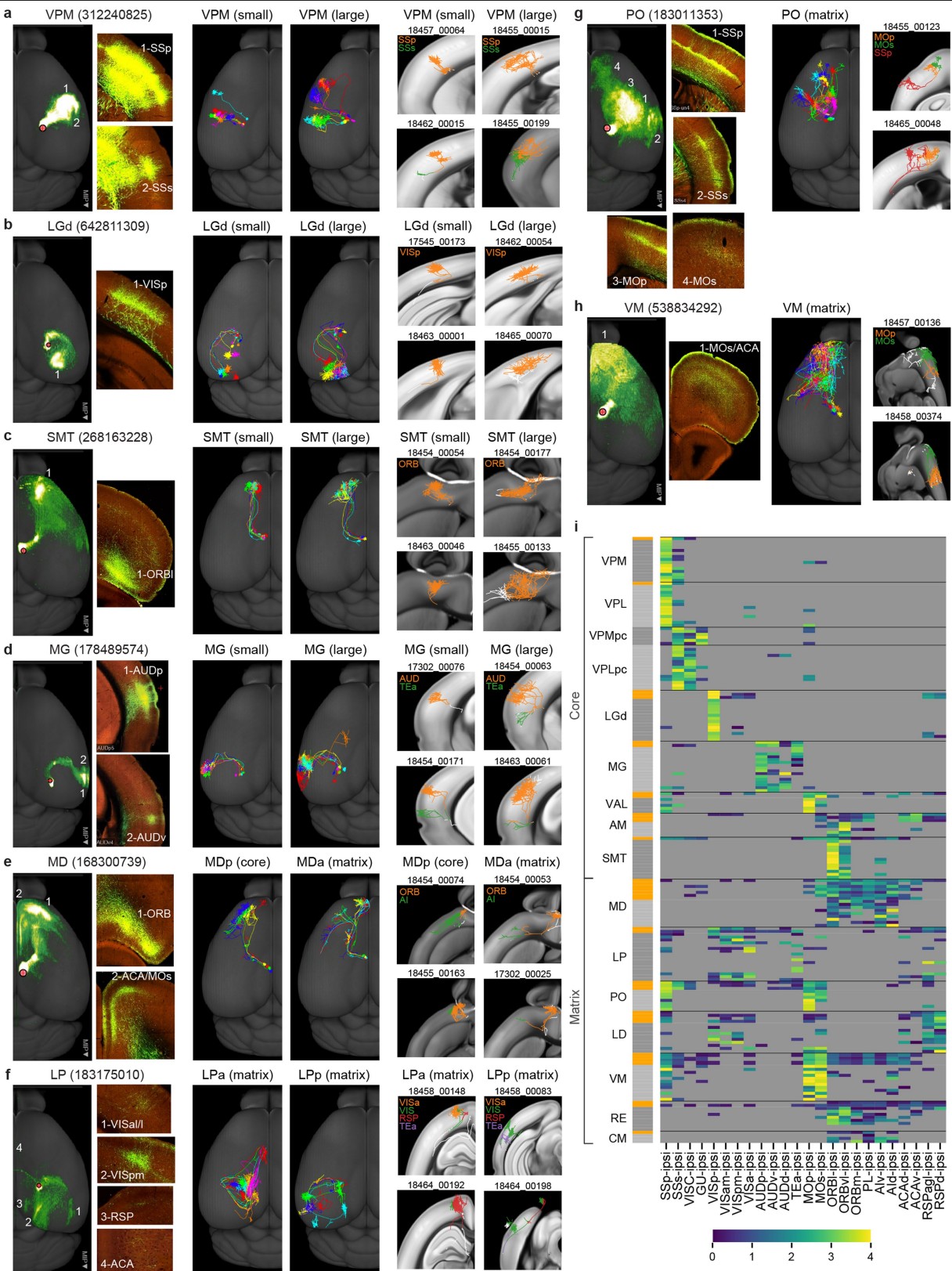

**Extended Data Fig. 12** | See next page for caption.

**Extended Data Fig. 12 | Long-range projection patterns of individual thalamic neurons in comparison with mesoscale population-level projections. a–h**, Axonal morphologies and projections of reconstructed single neurons compared with population projection patterns for thalamic nuclei VPM (**a**), LGd (**b**), SMT (**c**), MG (**d**), MD (**e**), LP (**f**), PO (**g**) and VM (**h**). For each nucleus: left panels, representative mesoscale experiments shown in a maximum projection whole-brain top-down view and individual higher-power images showing axon termination patterns in major target regions; middle panels, representative single neurons shown together in a whole-brain top-down view; right panels, each representative neuron is shown in a chosen plane to best capture the perpendicular (to pial surface) orientation of the main axon arbor with superimposed maximum projection view of the neuron's axon arbors. The chosen plane can be coronal (for **a**, **b**, **d**), horizontal (for **c**, **e**), sagittal (for **h**) or tilted (for **f**, **g**), based on the main cortical target region. Different cortical target regions are indicated by different colors. Small, small axon arbors. Large, large axon arbors. MDa and MDp, or LPa and LPp are the anterior and posterior parts of MD or LP respectively. **i**, Projection matrix showing comparison of thalamocortical projection patterns between mesoscale experiments and single neurons as well as among individual neurons, for each listed thalamic nucleus. Each row is a mesoscale experiment (labeled in orange in the left side bar) or a single cell (labeled in grey in the left side bar). Heatmap colors represent projection strengths, defined as ln(NPV × 100 + 1) for mesoscale experiments and ln(axon length) for single cells. Target regions are defined using thresholds of ln(NPV × 100 + 1) > 0.2 for mesoscale experiments and axon length > 1 mm for single cells. Regions below the thresholds are shown in grey. Overall, core-type thalamocortical neurons usually have one major axon arbor targeting L4 of the primary sensory or motor cortex of the corresponding modality, *i.e.*, VPM and VPL projecting to SSp, VPMpc to gustatory areas (GU), VPLpc to visceral area (VISC), LGd to primary visual area (VISp), MG to auditory areas (AUD), and VAL to MOp. Reconstructed neurons from AM, SMT and posterior MD also send a single major axon arbor to various parts of orbital cortex (ORB), with a similar mid-layer termination pattern, suggesting these neurons also belong to the "core" projection type. A small fraction of the core-type neurons (5.43% for VPM, 4.05% for VPL, 15.6% for MG, but 0% for LGd) have more than one axon arbor targeting different cortical areas. In the case of these cells in VPM and VPL, usually they have a larger main arbor targeting SSp, and a smaller secondary arbor targeting SSs (**a**). MG neurons with two or more cortical targets are mostly of the large-arbor type (**d**). These multi-target MG neurons are more like the matrix-type neurons, showing stronger projections to L1 and L5, located in the associational parts of MG (*e.g.*, MGm) medial to the core relay auditory nucleus, MGd and MGv. Single neurons are assigned to either small-arbor or large-arbor type (Extended Data Fig. 13). Large-arbor neurons account for 32.0% of the total reconstructions from VPM, 31.9% from VPL, 38.9% from LGd and 36.0% from MG. Neurons in SMT are also separable into small- and large-arbor types, whereas the current set of AM neurons all have small arbors and the posterior MD neurons all have large arbors. Matrix-type thalamocortical neurons exhibit a diverse range of projection and morphological patterns. For example, LP neurons preferentially project to two or more higher visual cortical areas. They do not directly project into VISp, 6 out of 16 reconstructed LP neurons has short axon fibers in VISp (average ~2 mm in VISp, substantially below the average of LGd neurons, ~26 mm). LP neurons can be roughly divided into an anterior and a posterior group, consistent with previous functional studies[71]. Posterior LP neurons mainly project to lateral and posterior higher visual areas, whereas anterior LP neurons mainly project to medial and anterior higher visual areas with some extending an axon projection into anterior cingulate cortex (ACA) (**f**). PO neurons project to both SSp and MOp/MOs. Their axon arbors in these target regions terminate broadly across layers with an apparent preference in L4 and lower L2/3, with 4 out of 7 sending rich axon arbors (>1 mm) to L1 (**g**). Neurons in anterior MD appear very different from those in posterior MD and are more similar to those in neighboring nuclei such as IAD and CM. They have multiple axon arbors that target multiple medial and lateral prefrontal cortical areas including prelimbic cortex (PL), ORB and agranular insular cortex (AI) (**e**). VM neurons have multiple axon arbors, heavily targeting MOp and/or MOs with additional branches targeting various somatosensory areas (**h**).

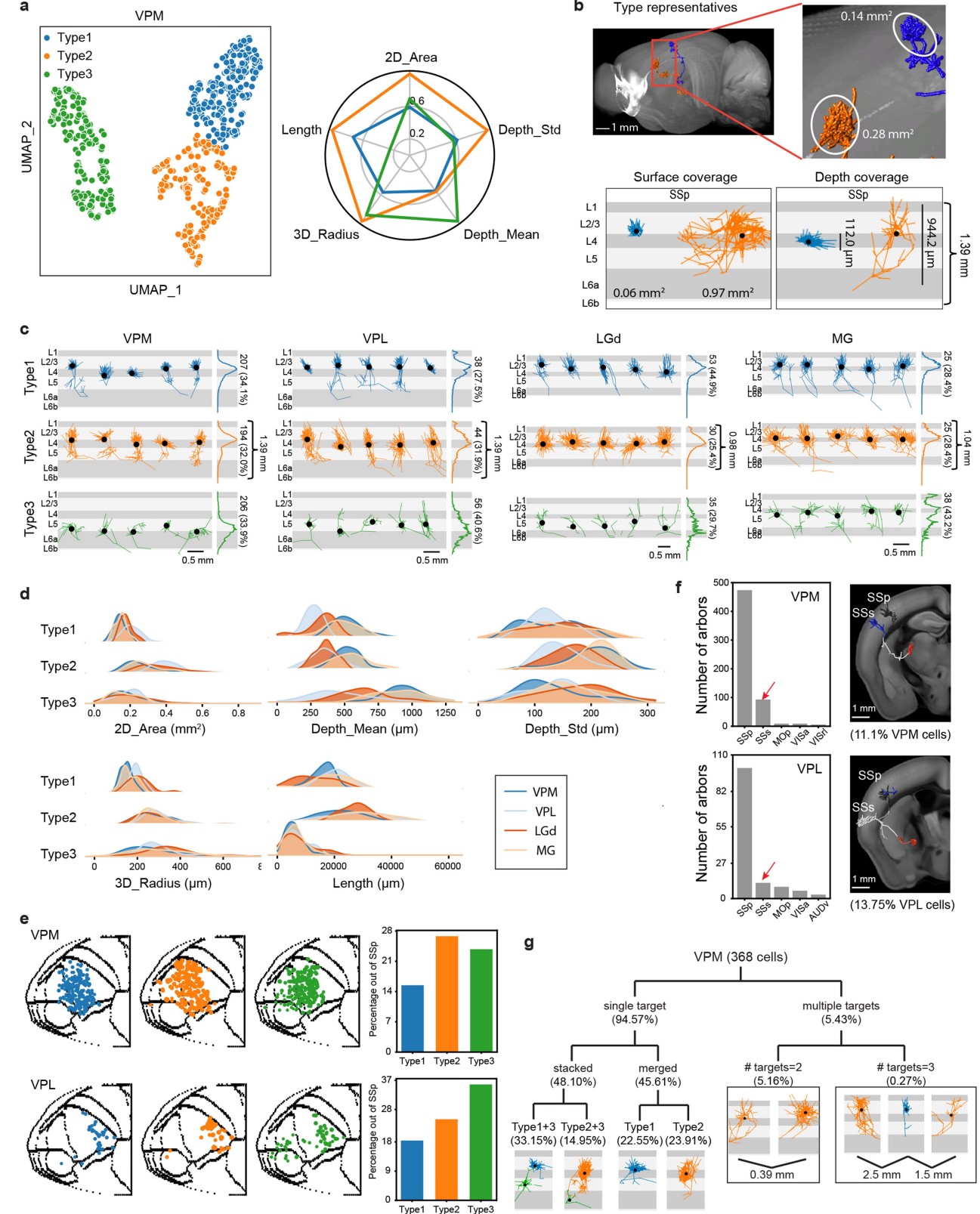

**Extended Data Fig. 13** | See next page for caption.

**Extended Data Fig. 13 | Thalamocortical axon arbor analysis.** Clustering of 944 cortical axon arbors from 586 neurons from VPM, VPL, LGd and MG reveal three types of arbors. Two major types of axon arbors target cortical layer 4 (spanning L4 and lower L2/3); a smaller 'type 1' arbor and a larger 'type 2' arbor (cortical area > 0.3 mm$^2$). The 'type 3' arbor terminate in cortical L6; this type is most often a minor collateral originating from the type 1 or type 2 arbor, so we did not use it to classify neurons. **a**, Clustering result indicates three types of cortical axon arbors in VPM neurons. Left, UMAP representation of VPM axon arbors. Right, polar plot of main features, values as normalized cluster averages. **b**, Representative (upper) and extreme (lower) examples of VPM cortical arbors. **c**, Examples grouped by thalamic nuclei and arbor types. In each sub-panel, vertical views are shown for 5 representative arbors, with branch length distribution for all neurons of the same cluster on the right side. Arbor number and percentage of the group are shown on the right side. **d**, Distribution of features grouped by thalamic nuclei and arbor types. **e**, Arbor locations of VPM and VPL neurons in 2D cortical map grouped by arbor types. Each dot represents the center of an arbor. Somas with small and large arbors are spatially intermingled in each nucleus. Right panels show percentage of arbors outside of the primary target of VPM or VPL neurons. **f**, (Left) Counts of VPM or VPL arbors in cortical regions. (Right) Examples of neurons with double arbors, one in SSp and the other in SSs. **g**, Variation of VPM neurons by arbor composition. 'Single target' neurons are described as 'stacked' or 'merged' by bi-layer or single-layer distribution. The stacked and merged groups can be further separated by arbor types. The 'multiple targets' group is divided by number of targets.

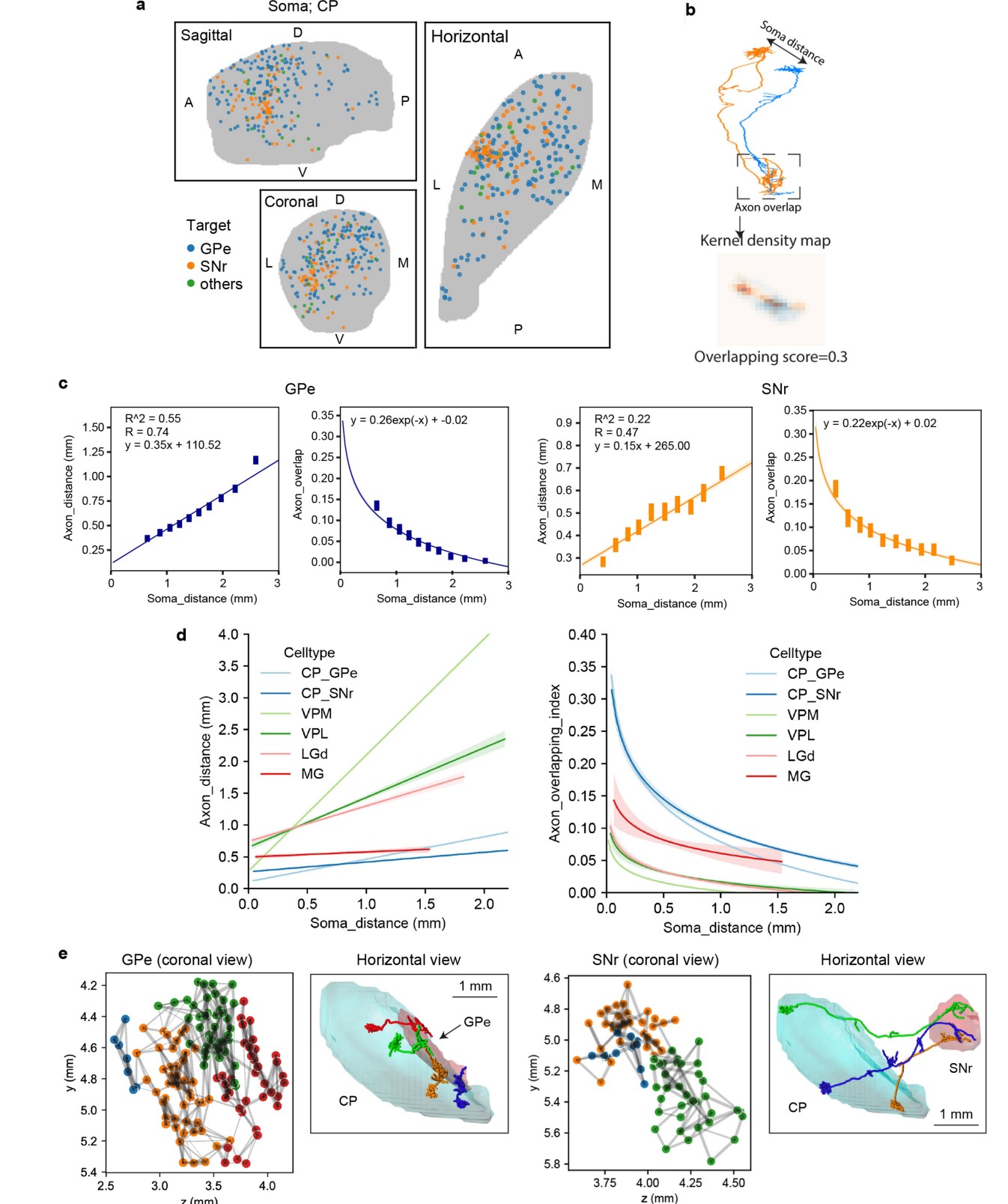

**Extended Data Fig. 14** | See next page for caption.

**Extended Data Fig. 14 | Striatal neuron morphologies. a**, Sagittal, coronal and horizontal views of soma distribution of CP neurons. Axes: D-V, dorsal to ventral; A-P, anterior to posterior; M-L, medial to lateral. We reconstructed 311 neurons in the dorsal striatum (CP) from 4 Cre driver lines: *Tnnt1*, *Plxnd1*, *Vipr2* and *Pvalb* (Supplementary Table 2). These neurons can be divided into 3 groups, largely intermingled with each other, based on their projection targets: those with main axon projections terminating in GPe (n = 180), SNr (n = 100) or within striatum itself (others, n = 31). **b**, Overlapping score of axons is calculated by estimating the kernel density map of individual axon arbors and the density-weighted average of overlapping areas for each arbor pair. **c**, Regression of distance between arbor centers (left panels) or overlapping score (right panels) in target regions (GPe or SNr) by soma distance. Linear and negative exponential models are used for distance and overlapping score, respectively. Vertical bars represent 95% confidence intervals of regression.

**d**, Comparison of arbor convergence across cell types. Regression curves are generated by the same approach as in **c**. Colors represent cell types. Center lines represent regression curves between soma distance and axon center distance (left panel), or soma distance and axon overlapping score (right panel). Light-shaded bands represent 95% confidence intervals. **e**, Clustering of axon overlapping by Louvain algorithm. Coronal views show axon arbor locations colored by clusters. Width of grey lines represents overlapping scores between arbor pairs. Horizontal views show example single neurons to illustrate topography of CP neuron projections. Cells are colored by cluster identities. In addition, the GPe-projecting type also has more elaborate axon arborization near the soma. Sholl analysis shows that the number of local crossings (< 1 mm to soma) of the GPe-projecting type is 2.9 times that of the SNr-projecting type.

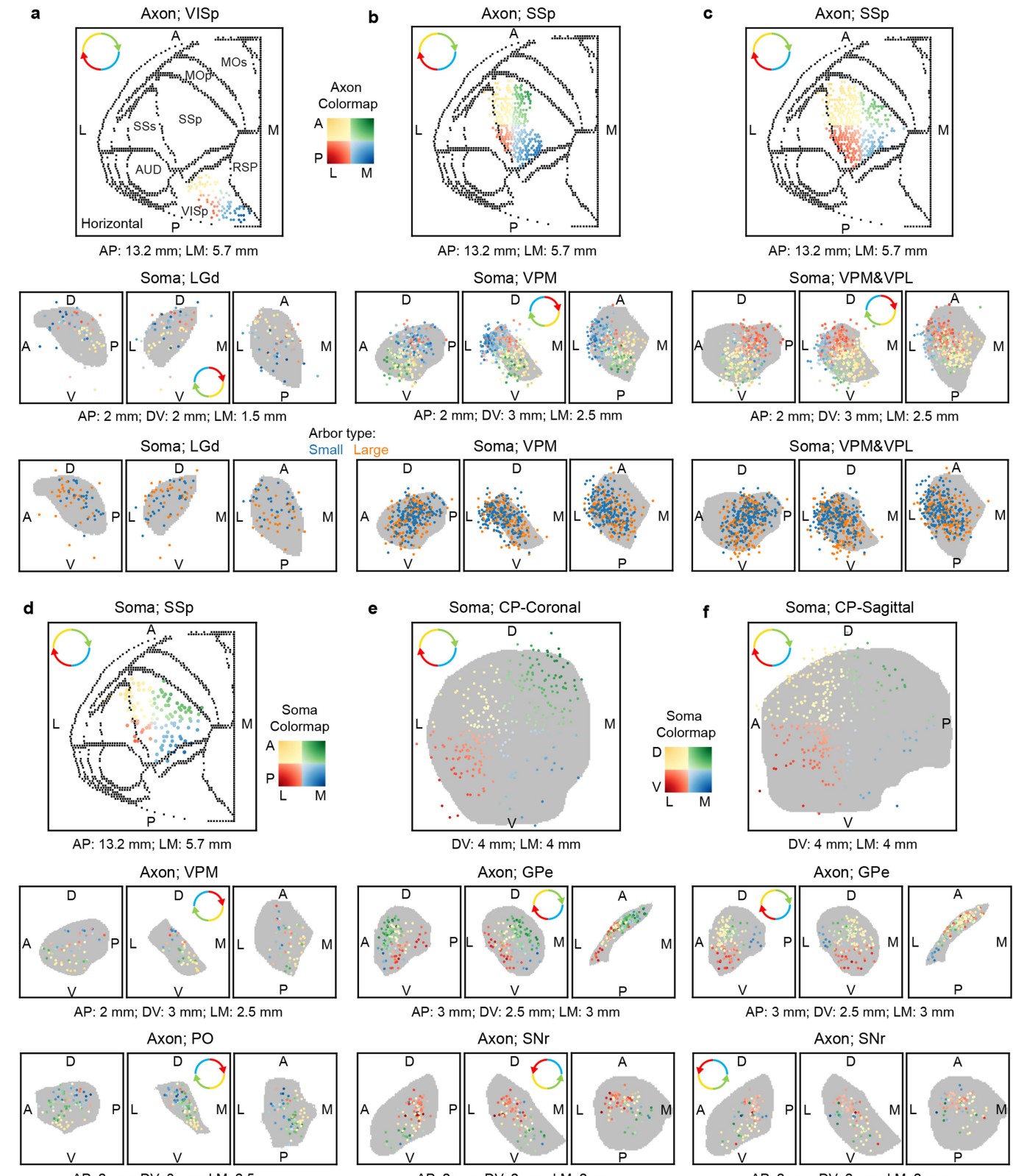

**Extended Data Fig. 15 |** See next page for caption.

**Extended Data Fig. 15 | Topography analysis. a–c**, Topographic distribution of the somas of LGd (**a**), VPM (**b**) and VPM&VPL (**c**) neurons and their terminal axon arbors in cortex. Top panels, axon arbors are shown in VISp (for LGd neurons) and SSp (for VPM and VPM&VPL neurons) in a cortical flatmap and divided into color-coded quadrants. Middle panels, corresponding soma locations labeled with the same color code are shown in LGd, VPM and VPM&VPL. Each color wheel with arrows denotes the observed general topographic orientation. Bottom panels, soma locations of neurons with small or large axon arbors. **d**, Topographic distribution of the somas of SSp L5 ET neurons and their terminal axon arbors in thalamus. Top panel, somas are shown in SSp in a cortical flatmap and divided into color-coded quadrants. Middle and bottom panels, corresponding axon arbor locations labeled with the same color code are shown in VPM and PO, respectively. **e**, **f**, Topographic distribution of the somas of CP neurons and their terminal axon arbors, shown in a coronal flatmap (**e**) or a sagittal flatmap (**f**). Top panels, somas are shown in CP in two projected planes, dorsoventral-mediolateral (**e**) and dorsoventral-anteroposterior (**f**), each divided into color-coded quadrants. Middle and bottom panels, corresponding axon arbor locations labeled with the same color code are shown in GPe (for GPe-projecting neurons) and SNr (for SNr-projecting neurons), respectively.

# nature research

# Reporting Summary

Nature Research wishes to improve the reproducibility of the work that we publish. This form provides structure for consistency and transparency in reporting. For further information on Nature Research policies, see our Editorial Policies and the Editorial Policy Checklist.

## Statistics

For all statistical analyses, confirm that the following items are present in the figure legend, table legend, main text, or Methods section.

| n/a | Confirmed | |
|---|---|---|
| ☐ | ☒ | The exact sample size (*n*) for each experimental group/condition, given as a discrete number and unit of measurement |
| ☐ | ☒ | A statement on whether measurements were taken from distinct samples or whether the same sample was measured repeatedly |
| ☐ | ☒ | The statistical test(s) used AND whether they are one- or two-sided<br>*Only common tests should be described solely by name; describe more complex techniques in the Methods section.* |
| ☐ | ☒ | A description of all covariates tested |
| ☒ | ☐ | A description of any assumptions or corrections, such as tests of normality and adjustment for multiple comparisons |
| ☐ | ☒ | A full description of the statistical parameters including central tendency (e.g. means) or other basic estimates (e.g. regression coefficient) AND variation (e.g. standard deviation) or associated estimates of uncertainty (e.g. confidence intervals) |
| ☐ | ☒ | For null hypothesis testing, the test statistic (e.g. *F*, *t*, *r*) with confidence intervals, effect sizes, degrees of freedom and *P* value noted<br>*Give P values as exact values whenever suitable.* |
| ☒ | ☐ | For Bayesian analysis, information on the choice of priors and Markov chain Monte Carlo settings |
| ☒ | ☐ | For hierarchical and complex designs, identification of the appropriate level for tests and full reporting of outcomes |
| ☒ | ☐ | Estimates of effect sizes (e.g. Cohen's *d*, Pearson's *r*), indicating how they were calculated |

*Our web collection on statistics for biologists contains articles on many of the points above.*

## Software and code

Policy information about availability of computer code

| | |
|---|---|
| Data collection | TissueCyte 1000, fMOST and Vaa3D were used in data acquisition processes |
| Data analysis | Vaa3D (version 3.604), TeraFly (version 2.5.101) and TeraVR (version bundled with Vaa3D) are available at Vaa3D's github site, https://github.com/Vaa3D, with both source code and binary executable. The mBrainAligner package is available via https://github.com/Vaa3D/vaa3d_tools/blob/master/hackathon/mBrainAligner.<br>Python package neuro_morpho_toolbox (https://github.com/pengxie-bioinfo/neuro_morpho_toolbox) is used for full morphology analysis. Custom data analysis notebooks are available via https://github.com/pengxie-bioinfo/BICCN_full_morphology.<br>STAR v2.5.3 and R GenomicAlignments package (RRID: SCR_018096) are used for RNA-seq alignment. R package scrattch.hicat (https://github.com/AllenInstitute/scrattch.hicat) is used for scRNA-seq analysis, including mapping retro-seq cells to reference taxonomy and re-clustering. |

For manuscripts utilizing custom algorithms or software that are central to the research but not yet described in published literature, software must be made available to editors and reviewers. We strongly encourage code deposition in a community repository (e.g. GitHub). See the Nature Research guidelines for submitting code & software for further information.

## Data

Policy information about availability of data

All manuscripts must include a data availability statement. This statement should provide the following information, where applicable:

- Accession codes, unique identifiers, or web links for publicly available datasets
- A list of figures that have associated raw data
- A description of any restrictions on data availability

Data availability statement is provided in the manuscript.

# Field-specific reporting

Please select the one below that is the best fit for your research. If you are not sure, read the appropriate sections before making your selection.

☒ Life sciences  ☐ Behavioural & social sciences  ☐ Ecological, evolutionary & environmental sciences

For a reference copy of the document with all sections, see nature.com/documents/nr-reporting-summary-flat.pdf

# Life sciences study design

All studies must disclose on these points even when the disclosure is negative.

| | |
|---|---|
| Sample size | No sample size calculation was performed. Sample sizes were determined by cell type specificity of mouse lines and limited by practical reasons, such as imaging quality and sparsity of labeling. Sample sizes are provided in Supplementary Tables 1 and 3. Specifically, 1-7 brains each transgenic mouse line used for generating neuron reconstructions for analysis in the current study had 3-7 whole-brain fMOST imaging datasets. Dozens to hundreds of cells were reconstructed from each selected brain. |
| Data exclusions | Full morphology dataset generated by this study has been made available. Low quality cells (weak labeling or too crowded labeling) that can't be fully reconstructed were excluded from reconstruction. For comparative analysis with mesoscale data, we excluded mesoscale experiments with injection contamination in neighboring brain regions.<br>All Retro-seq cells in the targeted regions were included for analysis. |
| Replication | Each mouse line used for analysis in the current study contained at least three mice for imaging data acquisition. For analysis of projection patterns, a minimum of three single cells was required for each cell type in each brain region. Each neuron reconstruction was validated by at least 2 or 3 annotators in the workflow. All attempts at replication were successful. |
| Randomization | This study did not involve allocation of experimental groups. Data were grouped by Cre-lines or soma locations. Samples were not randomized. |
| Blinding | Blinding is not applicable to the study design. There was no allocation of treatment and control groups. |

# Reporting for specific materials, systems and methods

We require information from authors about some types of materials, experimental systems and methods used in many studies. Here, indicate whether each material, system or method listed is relevant to your study. If you are not sure if a list item applies to your research, read the appropriate section before selecting a response.

## Materials & experimental systems

| n/a | Involved in the study |
|---|---|
| ☒ | ☐ Antibodies |
| ☒ | ☐ Eukaryotic cell lines |
| ☒ | ☐ Palaeontology and archaeology |
| ☐ | ☒ Animals and other organisms |
| ☒ | ☐ Human research participants |
| ☒ | ☐ Clinical data |
| ☒ | ☐ Dual use research of concern |

## Methods

| n/a | Involved in the study |
|---|---|
| ☒ | ☐ ChIP-seq |
| ☒ | ☐ Flow cytometry |
| ☒ | ☐ MRI-based neuroimaging |

## Animals and other organisms

Policy information about studies involving animals; ARRIVE guidelines recommended for reporting animal research

| | |
|---|---|
| Laboratory animals | We used transgenic mice that contain a combination of the following individual driver and reporter lines: Cux2-CreERT2, Fezf2-CreER, Gnb4-IRES2-CreERT2, Plxnd1-CreER, Pvalb-T2A-CreERT2, Tnnt1-IRES2-CreERT2, Vipr2-IRES2-Cre-neo, Snap25-IRES2-Cre, Slc17a7-IRES2-Cre, Esr2-IRES2-Cre, Ai139, Ai140, Ai82, Ai166, Ai14, Ai65F, and RCL-Sun1sfGFP. All transgenic mice were maintained in C57BL/6J congenic background. For each genotype of transgenic mice, we used both male and female mice, ages ranging from 8 weeks to 5 months old. |

Mice were housed in animal rooms on a 14/10 hr light/dark cycle (6am-8pm light). The room temperature was set at 70°F (21°C) and the relative humidity at 40%.

Wild animals

The study did not involve wild animals.

Field-collected samples

The study did not involve samples collected from the field.

Ethics oversight

All experimental procedures using live animals were performed according to protocols approved by Institutional Animal Care and Use Committee (IACUC) of the Allen Institute for Brain Science.

Note that full information on the approval of the study protocol must also be provided in the manuscript.

