## [Peer Review File · Nature]

Manuscript Title: Morphological diversity of single neurons in molecularly defined cell types

Reviewer Comments & Author Rebuttals

Reviewer Reports on the Initial Version:

Referees' comments:

Referee #1 (Remarks to the Author):

The manuscript „Brain-wide single neuron reconstruction reveals morphological diversity in molecularly defined striatal, thalamic, cortical and claustral neuron types“ by Peng et al. reports the labeling and whole-brain reconstruction of 1708 neurons from various regions of the mouse brain. The authors analyze the projection patterns at the single-neuron level and report a substantial morphological diversity even in molecularly “defined” neurons.

Embracing the complexity of neuronal projections, which repeatedly are defying clear molecular-to-connectivity correspondences, is of utmost importance. Even with substantial transcriptomic data available, a clear match between protein expression and axonal synaptic connectivity has not been found.

However, what is less clear to me is the novelty of the reported data. While this data represents a substantial amount of work, it is not exceptionally more than what was reported by the Janelia MouseLight project in 2019 (Winnubst et al., 2019 Cell; more than 1000 neurons reconstructed). Maybe I am missing a key point, but to me as a reader it is not clear what the major advance of the present data compared to Winnubst et al. is. The 70% more neurons are not clearly a leap forward. This is not to diminish the scale of work that went into the current project, but to me it is not clear that this dataset requires to be published in Nature rather than a more specialized journal or as a resource.

One could argue that the data on a defined molecular type and its diverse projection patterns constitutes some advance, but this analysis refers to one particular class of neurons (L6 Car3), and cannot be over-interpreted to allow the conclusion indicated in the title “morphological diversity in molecularly defined..”.

I am also underwhelmed by the presentation of the data – especially compared to Winnubst et al., which is much more succinct and understandable, the vast amount of overcrowded figures in the current manuscript does not help to understand the data better, nor focus on advances compared to the published state of the art.

Referee #3 (Remarks to the Author):

This manuscript „Brain-wide single neuron reconstruction reveals morphological diversity in molecularly defined striatal, thalamic, cortical and claustral neuron types“ by Hanchuan Peng and colleagues is a massive piece of work with heroic data collection regarding complete single neuron morphology reconstructions (including complete axons). The methods seem to match previous approaches published elsewhere (from Janelia Farm, including some of the authors here).

A recent landmark paper in Nature by Michael Economo and colleagues has shown that L5 cortical pyramidal neuron axons in the pyramidal tract divided into two distinct populations according to their RNA content that exquisitely followed a corresponding distinction of their axonal projections. That work has shown that axonal reconstructions are important to define distinctions into cell types that are to date not known. In contrast, in the work presently under review the authors find

a diversity of axonal projections even for neurons that group into genetically more similar groups. This most likely means, as the authors suggest (it „cannot presently be accounted for by transcriptomic subtypes“), that the transcriptome data is not sufficient to make any conclusions here (analysis is not precise enough).

The observed region-specific organizational rules of long-range axonal projections at the single cell level mentioned in the abstract remain relatively superficial (different neurons follow all kinds of different projections rules) resulting in a feeling of an overwhelming amount of data with a lack of novel insights after reading this manuscript.

The importance of this particular work therefore lies in my opinion more strongly in the novel pipeline for complete single cell reconstructions including transcriptome information. It includes a huge actual dataset that is now available for further analysis (data sharing here is a must). I therefore see this more as a methods paper than a research report and the novelty is a bit faded by the fact that none of the individual components of the pipeline seem particularly new and by the existing previous publication of different pipelines with similar resulting datasets.

The conclusion that full morphological characterization (including the extended axon) of the single cell is important is certainly true but is a well-accepted notion backed by a large body of literature.

Detailed comments

1) Most importantly, this paper needs to compare its pipeline with existing pipelines (e.g. Economo 2016 eLife/Winnubst 2019 Cell), the methods result in similar data and it is important to quantify the advantages and disadvantages of both.

2) The authors observe a relation between soma distance and axon distance / overlap. This seems to be a rather linear relation that maybe can be related to tracer studies that showed an exponential decay of connectivity between regions with their respective distance.

3) The juxtaposition between single neuron arborizations and mesoscale tracing results is striking and can be maybe elaborated on (quantified in more detail).

4) The presentation is a bit sloppy including the language. E.g. Figure 1c-h it is not clear what the different views are. L5ET seems to be missing a dendritic branch, what happened? Figure 2 panels are too small. Figure 3a legend does not match panel (where are the crosses?). Figure 3c are these long bars standard deviation?...

5) The paper feels a bit like a hybrid between an atlas and a herbarium but does not provide a consistent and novel organization of cell types by their axonal projections.

6) Some amazing topographic relationships are shown but not quantified in more detail and the underlying principles are not clear.

7) The first paragraph in the Introduction is true for cortex but not for example for more local axonal arborizations in various systems including for neurons of the central nervous system in many invertebrates.

Author Rebuttals to Initial Comments:

Referees' comments:

Overall Response:

We thank the referees for their constructive critiques, which have been extremely helpful in our effort to improve the manuscript. In the revised manuscript submitted here, we have attempted to address all the points raised by the reviewers and reorganized our manuscript substantially to simplify the results and highlight the major advances our study brings to the field. Here we provide a brief summary of the major changes we have made:

- Revised all figures and extended data figures, reorganized the order of the figures and Results text to better demonstrate the novel findings of the paper. In particular, added a new Fig 2 to provide an upfront summary of the key results of our systematic characterization of the complete morphologies of all the 11 major projection neuron types, to provide a clearer high-level narrative of our study before diving into the details.
- Added topography analysis of several cell types (new Fig 8f-h) as suggested by a referee, making this one of the major morphological/projectional features for cell type characterization. Removed many other figure panels to make the figures less cluttered with less essential analysis results.
- Added ~100 new reconstructions (mainly for L2/3, L5 IT and L5 ET neurons) to strengthen the comparative analysis of several cell types. In particular, added new analysis in Fig 5 to demonstrate the major differences among L2/3, L4 and L5 IT neurons. Also updated clustering analysis of L5 ET neurons, and we believe that the new clusters better capture the range of target-driven axon projection patterns.
- Also rechecked our entire dataset carefully and removed several dozens of duplicated reconstructions from different institutions. (These independently reconstructed duplicates are highly similar to each other, serving as another demonstration that our multi-site reconstruction teams have been generating high quality, (near) complete, and reproducible neuronal morphologies.) The net result is an increase of the total number of reconstructed neurons reported in the paper from 1708 to 1741, making it still the largest set of full neuronal morphologies reported to date.
- Revised Discussion extensively to clarify the novel insights generated from our study, especially on the morphological diversity at multiple levels – from distinction between cell types, to location-based variation within types, to individual cell variability, and the complex relationship between morphological diversity and transcriptomic profiles, which leads to the proposal that the multi-level morphological and projectional diversity of cell types and single cells may be the result of a combination of genetic programming and circuit interaction (in a potentially activity-dependent process).

Since the main critique from the referees is the lack of clear demonstration of novel advances obtained from our large-scale study, we have made a substantial effort to improve on this. Here we would like to first summarize what we believe are the major advances in both technical and conceptual fronts before going into the point-by-point responses to referees' individual comments. And we ask referees to read our revised Discussion to evaluate all the points we wish to make.

Our brain-wide full morphology technology platform and pipeline was established with funding from the BRAIN Initiative as one of the essential capabilities for comprehensive multimodal characterization of brain cell types in the BRAIN Initiative Cell Census Network (BICCN). The project involves a multi-team collaboration in close coordination with other cell typing efforts such as single-cell transcriptomics and anatomical tracing. We follow NIH mandates to make our data and tools associated with this project rapidly available to the community.

This manuscript is one of the core companion papers of the inaugural BICCN publication package, whose scope of work is described in its flagship paper, “A multimodal cell census and atlas of the mammalian primary motor cortex”, viewable as a bioRxiv preprint (<https://www.biorxiv.org/content/10.1101/2020.10.19.343129v1>). Our goal for this manuscript was to present a systematic characterization of the morphological and projectional features of multiple cell types across different regions, to derive generalizable rules underlying cell type specific morphological diversity. While doing this, we didn’t want to make a technical comparison with the MouseLight paper (Winnubst et al, Cell 2019) publicly even though we believed our approach has significant advantages over the published work. But by doing so we missed the opportunity to show to the referees the advances of our approach. So here we list out the advances for referees’ consideration. The key points listed here are also mentioned in various parts of the manuscript itself.

A) Technical merits and/or advances of our approach in comparison with a similar large-scale pipeline approach – the Janelia MouseLight platform:

- 1) MouseLight is a pipeline that has been contained within Janelia and inaccessible by the community. In contrast, our pipeline is modular and open. The work was already distributed across 5 institutions: fMOST imaging at HUST, CCF registration at Anhui U, and reconstruction done in three other sites (Allen Institute, SEU and WMU). Capabilities were built to allow flexible sharing of data and computational tools across sites, which will greatly facilitate sharing with the larger community.
- 2) Winnubst et al paper used pan-neuronal viral vector injection to label neurons, lacking cell type specific labeling. The cell types were inferred from the locations and morphologies of the neurons. The local sparsity of the labeling was also unclear, for example, we found that many of their reconstructed neurons were missing local axon arbors. In contrast, we use specially designed transgenic reporter mouse lines for robust and consistent sparse labeling and Cre driver lines for cell type specific targeting. Consequently, our reconstructions are more complete (including both local and long-range axon arbors), and cell type specific labeling is particularly significant as it allows us to examine the relationship between transcriptomically defined cell types and morphology/projection patterns. Even though the individual mouse lines were recently published while we were preparing the manuscript, our work in finding the unique combination of them and the optimal sparse labeling conditions is described for the first time in this manuscript.
- 3) We further included two sets of single-cell Retro-seq data to corroborate our findings. These are novel, unpublished datasets. They were mapped to a novel, unpublished transcriptomic taxonomy that covers all the cortical regions studied in this manuscript. This taxonomy is reported in our paper (Yao et al 2020) titled “A taxonomy of transcriptomic cell types across the isocortex and hippocampal formation”, in press in Cell now and to be published online in a few weeks, also viewable as a bioRxiv preprint (<https://www.biorxiv.org/content/10.1101/2020.03.30.015214v1>). We point this out in case the referees were not aware of this work.
- 4) Our manuscript here presented reconstructed morphologies of a different and more expanded set of cell types than the Winnubst et al paper. Winnubst et al had neurons from zona incerta, subiculum, motor cortex [IT, PT (called ET in the BICCN publication package, see flagship paper for rationale) and CT cells], and 4 thalamic nuclei. In comparison, our paper has neurons from striatum, ~20 thalamic nuclei, IT and ET neurons from a number of cortical areas (analysis done on those cells from 4 areas), L6 Car3 cells throughout cortex, and claustrum neurons. From this larger and different set of cell types, we are able to find new morphological types and derive new insights and general rules. Our paper is also unique in the BICCN publication package as it goes much beyond the focus on primary motor cortex

as described in the flagship paper. Our coverage of multiple cell types in multiple brain regions allows the identification of more generalizable patterns and rules.

- 5) Furthermore, our pipeline is rapidly expandable beyond the scope of this manuscript. The fMOST team recently published a new platform named HD-fMOST (Zhong et al, Nature Methods 2021), which has higher throughput and lower background and allows even more efficient data generation. The number of brain datasets suitable for reconstruction has already grown to >130, much beyond what is reported in this manuscript (53). The number of reconstructed neurons we have has also surpassed 2,500, although they cannot be all included in this manuscript due to the time needed to check and analyze them. We make all our raw and processed image data as well as our computational tools publicly available, to enable community-based reconstruction effort to utilize this valuable resource.

B) Our approach and our data allowed us to obtain new insights which are described in more detail in the revised Discussion section. Here we provide a brief summary:

- 1) We systematically examined multiple levels of morphological properties and their correspondence with transcriptomic profiles. At the highest level, there is a high degree of concordance between major transcriptomic and projection neuron types. Neurons belonging to different transcriptomic subclasses and major types have highly distinct morphological and projectional properties.
- 2) The morphological distinction among major projection neuron types is reflected not only in the specific set of projection targets each type has, but also the total number of targets and various characteristics of each type's projection pattern (e.g., convergence vs divergence, feedforward vs feedback), all of which are likely related to the specific function of each type. Thus, cell type-specific morphological features can bridge the molecular genetic identity of a cell type and its function.
- 3) At the intermediate level, within each projection neuron type, neurons follow region-specific and topographic organizational rules. Region-specific features are seen in all cortical and thalamic projection types containing neurons from different subregions. Topographic correspondence between soma locations and major terminal axons is seen in various orientations in all projection types examined. The discrete region-specific morphological features are in contrast with but may arise from the continuous transcriptomic variations between neighboring regions.
- 4) At the lowest, single cell level, the degree of similarity or variability between individual neurons within a given type also varies across types. Neuron types with higher numbers of targets and more divergent projection patterns have higher individual variability with each neuron selecting only a subset of the targets. The selection of a subset of targets appears stochastic in some cases but can also allow further clustering of individual cells into target-driven subtypes. Retro-seq data does not show clear correlation of target specificity at this level with currently defined transcriptomic clusters.
- 5) Overall, our systematic and comprehensive characterization reveals morphological diversity at multiple levels – from distinction between cell types, to location-based variation within types, to individual cell variability, and the complex relationship between morphological diversity and transcriptomic profiles across these levels. Morphological features at high levels are closely related to the neuron's molecular identities, whereas those at more refined levels may underlie the specific functional roles each neuron plays in the circuit it is embedded in. Our study suggests that the multi-level morphological and projectional diversity of cell types and single cells may be the result of a combination of genetic programming and circuit interaction (in a potentially activity-dependent manner) during development.

Collectively, we believe these findings from multiple cell types across multiple regions do provide a consistent and novel organization of cell types by their axonal projections, going much beyond the individual findings from a single cell type and/or a single region in previous studies. To the best of our knowledge, this kind of systematic study anchored on the concept of cell types using such a large-scale single cell complete morphology dataset had not been done before. Thus, we believe our work is unprecedented in both scope and depth and does represent a major advance in the cell types field. Our work also showcases a unique and critical component of the cell type studies encompassed in the BICCN publication package, together providing a roadmap for future studies in uncovering cell type-based organizational principles in the brain.

Our detailed point-by-point responses are shown below. We hope the referees agree that we have revised our manuscript satisfactorily for publication in Nature. We would be happy to address any further comments they may have.

Referee #1 (Remarks to the Author):

The manuscript „Brain-wide single neuron reconstruction reveals morphological diversity in molecularly defined striatal, thalamic, cortical and claustral neuron types“ by Peng et al. reports the labeling and whole-brain reconstruction of 1708 neurons from various regions of the mouse brain. The authors analyze the projection patterns at the single-neuron level and report a substantial morphological diversity even in molecularly “defined” neurons.

Embracing the complexity of neuronal projections, which repeatedly are defying clear molecular-to-connectivity correspondences, is of utmost importance. Even with substantial transcriptomic data available, a clear match between protein expression and axonal synaptic connectivity has not been found.

We fully agree with this referee on the importance of resolving the molecular-to-connectivity correspondence issue. What we meant by molecularly “defined” neurons is that each neuron belongs to a transcriptomically defined subclass that is usually labeled by a subclass-selective Cre driver line.

However, what is less clear to me is the novelty of the reported data. While this data represents a substantial amount of work, it is not exceptionally more than what was reported by the Janelia MouseLight project in 2019 (Winnubst et al., 2019 Cell; more than 1000 neurons reconstructed). Maybe I am missing a key point, but to me as a reader it is not clear what the major advance of the present data compared to Winnubst et al. is. The 70% more neurons are not clearly a leap forward. This is not to diminish the scale of work that went into the current project, but to me it is not clear that this dataset requires to be published in Nature rather than a more specialized journal or as a resource.

We recognize that we did not do a good job in demonstrating the novelty and major advances of our work in our original manuscript. We have now made substantial effort to improve on this point. Our morphological characterization is much more comprehensive and in-depth compared to previous studies, in particular Winnubst et al, and we strongly believe that it does lead to major advances. The major advances are in two areas – technical and conceptual, as summarized in our Overall Response above.

One could argue that the data on a defined molecular type and its diverse projection patterns constitutes some advance, but this analysis refers to one particular class of neurons (L6 Car3), and cannot be over-interpreted to allow the conclusion indicated in the title “morphological diversity in molecularly defined..”.

The referee is correct that the main advances of our study indeed include the demonstration of diverse projection patterns from defined molecular types. However, we did not just show this in one type of neurons (L6 Car3), but in 11 major neuron types from several brain regions (cortex, claustrum, thalamus and striatum). Beyond a description of the diversity, we further derived what we believe are the general rules governing the morphological and projectional diversity, which include molecular correspondence, divergent or convergent projection, axon termination pattern, regional specificity, topography, and individual variability (see new Fig 2). Importantly, we demonstrate that these rules are manifested in a variety of ways in different types of neurons from different brain regions, likely related to the specific functions different cell types play in their respective circuits.

I am also underwhelmed by the presentation of the data – especially compared to Winnubst et al., which is much more succinct and understandable, the vast amount of overcrowded figures in the current manuscript does not help to understand the data better, nor focus on advances compared to the published state of the art.

We thank the referee for pointing out this issue. We have now reorganized and simplified all the figures. We removed some detailed analyses that are not essential to the main conclusions. We put two summary figures (new Figs 2 and 3) in the front to more clearly show the insights gained from our systematic study before going into details.

Referee #3 (Remarks to the Author):

This manuscript „Brain-wide single neuron reconstruction reveals morphological diversity in molecularly defined striatal, thalamic, cortical and claustral neuron types“ by Hanchuan Peng and

colleagues is a massive piece of work with heroic data collection regarding complete single neuron morphology reconstructions (including complete axons). The methods seem to match previous approaches published elsewhere (from Janelia Farm, including some of the authors here).

We thank the referee for recognizing the massive and heroic data collection our multi-site collaborative teams carried out. We would like to point out that our approach not only matches the Janelia MouseLight approach but also has some significant advances, as laid out in the Overall Response above as well as our response to other comments from this referee below.

A recent landmark paper in Nature by Michael Economo and colleagues has shown that L5 cortical pyramidal neuron axons in the pyramidal tract divided into two distinct populations according to their RNA content that exquisitely followed a corresponding distinction of their axonal projections. That work has shown that axonal reconstructions are important to define distinctions into cell types that are to date not known. In contrast, in the work presently under review the authors find a diversity of axonal projections even for neurons that group into genetically more similar groups. This most likely means, as the authors suggest (it „cannot presently be accounted for by transcriptomic subtypes“), that the transcriptome data is not sufficient to make any conclusions here (analysis is not precise enough).

As several of us from Allen Institute (Zeng, Tasic, et al) are main collaborators and co-authors on the Economo et al Nature 2018 paper, we are very familiar with that work. We believe that the outcome of our current study is not in contrast with that previous work but extending much beyond it. The Economo et al study examined the morphology and in vivo physiology of two projection types within one subclass of neurons, L5 ET, in one cortical region (ALM), and found that the two morphologically defined projection types correspond to distinct transcriptomic types within the L5 ET subclass very well.

We disagree with the notion that only a positive correspondence result is interesting; otherwise the result is uninteresting or inconclusive. We realize that it is difficult to prove a negative result is valid; however, we would also like to point out that even from a randomly variable gene expression dataset one can always find some genes correlated with a certain phenotype, so one cannot simply ascertain that a positive result is valid either. Further studies to demonstrate causality is necessary even in the positive correlation case. We also don't think that we didn't try hard enough. We used the same approaches (e.g. complete morphology reconstruction and Retro-seq) as in the Economo study. The distinction between the two L5 ET types was very obvious in the Economo study, just like the distinction among the multiple major cell types we uncovered here. We have actually gone a step beyond the Economo study and further characterized other aspects of morphological diversity to provide a more comprehensive view. However, we acknowledge that we may not have presented our conclusions clearly in the original version of the manuscript.

As now summarized in the new Fig 2 and revised Discussion, our current study examined neurons from 8 transcriptomic subclasses in multiple cortical areas and 3 subcortical regions, and revealed a more comprehensive landscape of morphological and projectional diversity as well as a more complex relationship between morphology and transcriptome. Consistent with the Economo et al study, we found that major projection types are highly distinct from each other and they largely correspond to transcriptomic subclasses and major types. At the same time, we identified further morphological diversity within each major projection type at multiple levels – convergent or divergent projections, differential axon termination patterns, regional specificity, topography, and variability among single cells. We further incorporated two Retro-seq experiments showing that in both IT and Car3 subclasses, neurons projecting to different targets are not distinguishable by their leaf-node transcriptomic cluster identity. Thus, compared to the Economo et al study, we believe that our investigation of the correspondence issue is more thorough rather than “not precise enough”, and a negative conclusion is nevertheless a solid conclusion.

We believe that the findings and conclusions we have made as outlined in the Overall Response above, hopefully better presented now in the revised manuscript, provide a more realistic and accurate picture of the relationship between transcriptomic and morphological diversity, which is important for the field to be informed of and highlights the need for new hypotheses. This more nuanced picture is also consistent with recent studies from our other companion papers in the same BICCN publication package, from Xiaowei Zhuang’s lab (<https://www.biorxiv.org/content/10.1101/2020.06.04.105700v1>) and Ed Callaway’s lab (<https://www.biorxiv.org/content/10.1101/2020.04.01.019612v2>). On the other hand, we do agree that further investigation of this issue using more sophisticated approaches is needed, especially when a neuron type has multiple targets and each neuron within that type can choose a variable subset of targets.

The observed region-specific organizational rules of long-range axonal projections at the single cell level mentioned in the abstract remain relatively superficial (different neurons follow all kinds of different projections rules) resulting in a feeling of an overwhelming amount of data with a lack of novel insights after reading this manuscript.

We thank the referee for this critical feedback. While we respectfully disagree with the notion that our results are superficial, we recognize that we did not do a good job in clearly articulating our points.

As a major component of BICCN’s comprehensive efforts in multi-modal characterization of brain cell types, we built a pipeline to generate a large number of high-resolution whole-brain fluorescent image datasets and systematically reconstruct complete single neuron morphologies from many cell

types throughout the brain. The goal is to understand organizing rules of cell type diversity at morphological level, an essential component of cell type characterization, and compare the uncovered rules with those derived from other types of characterization (e.g., single-cell transcriptomics) in order to arrive at a universally applicable and proper level of definition of cell types.

Our study uncovered different kinds of rules for different cell types in different regions. We realize that this may be confusing to readers. Thus, in the revised manuscript, we have created a new Fig 2 to categorize the rules and compare how these rules are applied in different cell types. We have also extensively reorganized and revised the Results and Discussion sections to clarify the new insights obtained in our study (as outlined in the above Overall Response).

Our investigation of the correspondence between morphology and transcriptomics results in a major conclusion on the relationship between the two, which is that transcriptomic subclasses and major types do have highly distinct morphological and projectional properties, however, within each major type (which has consistent transcriptomic and morphological properties), there is additional morphological diversity in regionality and variability that cannot be accounted for by the preexisting transcriptomic clusters. For example, projection patterns are region specific whereas transcriptomic clusters/types are often shared among regions (e.g., among cortical regions, among thalamic nuclei, and between cortex and claustrum). We believe this is an important conclusion by itself, and we propose several different mechanisms that may explain the origin of the morphological diversity, namely, molecular instructing signatures during development only, activity-dependent mechanisms, or simply stochasticity, all of which may be utilized under the same or different situations. The latter two mechanisms do not require transcriptomic correspondence, but it doesn't mean that they are less important.

The importance of this particular work therefore lies in my opinion more strongly in the novel pipeline for complete single cell reconstructions including transcriptome information. It includes a huge actual dataset that is now available for further analysis (data sharing here is a must). I therefore see this more as a methods paper than a research report and the novelty is a bit faded by the fact that none of the individual components of the pipeline seem particularly new and by the existing previous publication of different pipelines with similar resulting datasets.

We agree that the unique pipeline we have built is another important aspect of our work. Even though some components of the pipeline were published before, there are a number of advantages as outlined in the Overall Response above which we will not repeat here. Our pipeline is highly productive, in addition to the data included in this manuscript, we have now generated >180 fMOST whole-brain datasets and reconstructed >2,500 neurons.

The conclusion that full morphological characterization (including the extended axon) of the single cell is important is certainly true but is a well-accepted notion backed by a large body of literature.

Our work indeed reinforces the well-recognized importance of full morphological characterization including the complete axons, but we also demonstrate with extensive data and analysis how such characterization will reveal special properties of different cell types and circuits which will have significant functional implications. Our work also sends an important message to the community that transcriptomics alone is not sufficient to define cell types and circuit properties, especially in regionality, topography and variability, and advocates a holistic approach in defining cell types and how to reconcile the differential variations between different modalities (e.g. transcriptomics and morphology in this case) at these more refined but functionally relevant levels.

Detailed comments

1) Most importantly, this paper needs to compare its pipeline with existing pipelines (e.g. Economo 2016 eLife/Winnubst 2019 Cell), the methods result in similar data and it is important to quantify the advantages and disadvantages of both.

Point-by-point comparison between our pipeline and the Janelia MouseLight pipeline is provided in the Overall Response above. Main advantages of our pipeline are its open, modular, flexible and collaborative operation, robust and consistent cell type-selective sparse labeling, high efficiency data generation, and timely public release of data prior to publication. Regardless of advantages and disadvantages, the field definitely needs more than just one or two pipelines.

2) The authors observe a relation between soma distance and axon distance / overlap. This seems to be a rather linear relation that maybe can be related to tracer studies that showed an exponential decay of connectivity between regions with their respective distance.

We think these may be two separate issues. Our investigation of the terminal axon arborization distance/overlap is for the purpose of evaluating convergent projections. We found that striatal medium spiny neurons (MSNs) have a higher degree of terminal axon overlap/convergence in target regions such as GPe and SNr, compared to thalamic core cells. We believe what the referee refers to here is the interconnectivity between two regions decrease exponentially with increasing distance between the two regions.

3) The juxtaposition between single neuron arborizations and mesoscale tracing results is striking and can be maybe elaborated on (quantified in more detail).

We have provided juxtaposed comparison between single neuron morphology and population level mesoscale tracing for every cell type included in this study. A quantitative comparison was shown in the original Fig 8 which we have now moved it up to Fig 3, to make the quantitative comparison result more prominent. We found that single cells of each type combined together can recapitulate the population level mesoscale projection pattern well for majority of the cell types and regions, except for some thalamic matrix-type nuclei. The poor correspondence in these matrix nuclei may be due to the following reasons: mesoscale experiments may have contaminating labeling of neurons from nearby nuclei, the numbers of reconstructed neurons for these nuclei are too small, or the reconstructed neurons may represent only a subset of the cell types located in these nuclei.

4) The presentation is a bit sloppy including the language. E.g. Figure 1c-h it is not clear what the different views are. L5ET seems to be missing a dendritic branch, what happened? Figure 2 panels are too small. Figure 3a legend does not match panel (where are the crosses?). Figure 3c are these long bars standard deviation?...

We are sorry about the sloppy appearance of the figures. We have now substantially revised all the figures, removed a number of non-essential analyses, corrected others and simplified all the figures overall.

Specifically, the original Fig 1c-h were images taken directly from the fMOST image datasets. Each fMOST section is only 1 um thick which defines the Z-resolution of the dataset and there are >10,000 sections (image planes) per brain. To provide a simpler overview of each dataset, a max intensity projection (MIP) dataset was generated by collapsing every 100 consecutive 1-um sections into a single plane. Thus, each MIP image contains 100-um thick tissue. The images shown in Fig 1c-h were taken from the MIP image series directly, from different brain regions as the labels indicated, without rotating the images. We use these images to demonstrate the robust fluorescent signals from a variety of neuron types that clearly visualize the very fine axon fibers and even boutons. One L5 ET cell shown is missing a dendritic branch because the image is only 100 um thick. In any case, we have now moved the example fMOST images in the original Fig 1c-h to Extended Data Fig 1, combined with images from additional cell types in the original Ext Data Fig 2, and reorganized the entire figure.

We moved the original Fig 2 panel a to new Fig 1 panel c. This is an illustration of our data processing and analysis pipeline. The individual image panels show the many different components of the pipeline; they do not convey specific scientific information, so we kept them small.

The original Fig 3 now becomes part of new Fig 8. We removed the Cre line labels which include the crosses in panel a because they do not convey essential information. The vertical bars in panel c represent 95% confidence intervals of regression. We have now added this note into the figure legend.

5) The paper feels a bit like a hybrid between an atlas and a herbarium but does not provide a consistent and novel organization of cell types by their axonal projections.

We thank the referee for this excellent point. We have now generated a new Fig 2 to illustrate the consistent and novel organization of cell types by their axonal projections.

6) Some amazing topographic relationships are shown but not quantified in more detail and the underlying principles are not clear.

We thank the referee for this excellent point. We have now included a visualization of topographic relationships of some cell types in the second half of the new Fig 8. The results reveal interesting corresponding topographical orientations in thalamus nuclei and cortical regions, as well as in striatum and its target regions GPe and SNr, at a coarse level. We are not sure what additional quantitative analyses may be helpful to uncover what kind of underlying principles. We are also hesitant to do detailed quantification due to the inherently imperfect registration of the individual fMOST datasets to the common coordinate framework (CCF).

7) The first paragraph in the Introduction is true for cortex but not for example for more local axonal arborizations in various systems including for neurons of the central nervous system in many invertebrates.

We thank the referee for pointing out this oversight. We have now added the mentioning of *Drosophila* work into the first paragraph of Introduction.

Reviewer Reports on the First Revision:

Referees' comments:

Referee #1 (Remarks to the Author):

The revised manuscript "Brain-wide single neuron reconstruction reveals morphological diversity in molecularly defined striatal, thalamic, cortical and claustral neuron types" by Peng et al is a substantial improvement compared to the initial version. The authors have followed our advice to focus the presentation, clarify the distinction to previous work, in particular the Janelia effort, and emphasize the finding of projectional diversity among transcriptomically defined neuron types. These are now well understandable already in figure 2 and explicitly in Fig. 7 and associated text and are from this reviewers' perspective a very important piece of data. This is in particular true in the context of attempts to use transcriptomic definitions as an exclusive source of neuronal type definitions, see for example the recent nomenclature proposals. These rely on the assumption that a transcriptomic identity is all it takes to infer the circuitry of neurons and the data presented here -if taken at face value- would clearly refute this at least for some of the transcriptomic classes.

Without asking for too much two concerns remain. I am not suggesting that they need to be addressed with novel data but just want to put them out here for potential further discussion. One is how sure can we be that the axonal projection diversity among a common transcriptomic class, in particular for the cortical neurons, is not affected by variability in reconstruction success. In other words: If based on missed side-branches axonal projections were to differ for these methodological reasons then the conclusion would of course be strongly affected by that. I am sure the authors have these methodological controls available so I suggest to make them very prominent in this context.

Secondly we could of course wonder whether the chosen transcriptomic identity in this particular case, particularly the Car3 neurons could be further subdivided at the transcriptomic level to capture projection differences. This of course could be a continuous definition loop so I am not suggesting that this needs to be analyzed with novel data - but it could potentially be commented on whether that is what the authors think we are looking at.

Referee #3 (Remarks to the Author):

Thank you, the revision addressed my concerns.

In the course of the revision, it became clearer that the major claim of this paper is that cell types that are currently indistinguishable using transcriptomics may exist that have differing axonal projections. A number of these cell types are characterized here, which is important for our principled understanding of how the brain is organized. This manuscript remains in my opinion predominantly a methods paper that proposes a new pipeline and offers amazing new data. It is essential here that the data are shared upon publication.

This manuscript is one step away from the Cell paper by Winnubst et al (including here transcriptomics information). However, given the methodological feat here, this work is a huge achievement and the result of a large collaborative effort. The new insights however could be more significant with a bit of effort from the authors. Clearly, the authors did not think that it is necessary in this work to analyze their data further (see their responses to the reviewers). More insights will most likely come from the subsequent usage of this dataset.

Author Rebuttals to First Revision:

Final response to referees

Referee #1 (Remarks to the Author):

The revised manuscript "Brain-wide single neuron reconstruction reveals morphological diversity in molecularly defined striatal, thalamic, cortical and claustral neuron types" by Peng et al is a substantial improvement compared to the initial version. The authors have followed our advice to focus the presentation, clarify the distinction to previous work, in particular the Janelia effort, and emphasize the finding of projectional diversity among transcriptomically defined neuron types. These are now well understandable already in figure 2 and explicitly in Fig. 7 and associated text and are from this reviewers' perspective a very important piece of data. This is in particular true in the context of attempts to use transcriptomic definitions as an exclusive source of neuronal type definitions, see for example the recent nomenclature proposals. These rely on the assumption that a transcriptomic identity is all it takes to infer the circuitry of neurons and the data presented here -if taken at face value- would clearly refute this at least for some of the transcriptomic classes.

We thank the referee for recognizing the significant improvements we have made to the paper. Indeed, transcriptomic definitions alone, especially from unsupervised clustering without incorporating other types of cellular properties, may not fully capture the nature of cell type diversity and specificity. The morphologies and projection patterns of neurons suggest additional mechanisms leading to such diversity and specificity as we have attempted to systematically document in our manuscript.

Without asking for too much two concerns remain. I am not suggesting that they need to be addressed with novel data but just want to put them out here for potential further discussion. One is how sure can we be that the axonal projection diversity among a common transcriptomic class, in particular for the cortical neurons, is not affected by variability in reconstruction success. In other words: If based on missed side-branches axonal projections were to differ for these methodological reasons then the conclusion would of course be strongly affected by that. I am sure the authors have these methodological controls available so I suggest to make them very prominent in this context.

We agree with the referee that high-quality complete reconstructions with no or minimal missing branches is of utmost importance to ensure that the results and conclusions of our study are solid. That is why we established a set of stringent QC processes as shown in Extended Data Figure 4 and in Methods. We adopted the same completeness assessment criterion as the Janelia MouseLight project, that is, each axon branch should end at an enlarged fluorescent spot (i.e., a supposedly bouton) rather than taper off. We routinely reconstructed multiple neurons of the same subclass from different regions in the same mouse brain where labeling and imaging qualities are consistent across individual neurons, and the morphological/projectional difference among these neurons is unambiguous to us.

Secondly we could of course wonder whether the chosen transcriptomic identity in this particular case, particularly the Car3 neurons could be further subdivided at the transcriptomic level to capture projection differences. This of course could be a continuous definition loop so I am not suggesting that this needs to be analyzed with novel data - but it could potentially be commented on whether that is what the authors think we are looking at.

We did try to further subdivide the Car3 neurons by re-clustering of our transcriptomes, as shown in Extended Data Figure 11. The cortical and claustral Car3 neurons with different projection targets were still not segregated in the more refined set of clusters. In the future, supervised clustering can be done to identify transcriptomic signatures that correlate with projection differences as we also mention in Discussion, however, as the referee points out here it might be a circular argument that will need to be tested and validated in perturbation types of experiments.

Referee #3 (Remarks to the Author):

Thank you, the revision addressed my concerns.

In the course of the revision, it became clearer that the major claim of this paper is that cell types that are currently indistinguishable using transcriptomics may exist that have differing axonal projections. A number of these cell types are characterized here, which is important for our principled understanding of how the brain is organized. This manuscript remains in my opinion predominantly a methods paper that proposes a new pipeline and offers amazing new data. It is essential here that the data are shared upon publication.

We thank the referee for recognizing the importance of our findings in furthering our principled understanding of how the brain is organized. This is indeed a very large-scale study for which we can only present the initial level of analyses in this manuscript. There is much to learn from our datasets, by both analyzing the neuron reconstructions in greater detail using additional approaches and reconstructing more neurons from the fMOST image data. Thus, we have made all data and computational tools publicly available, as shown in the Data Availability and Tool Availability sections.

This manuscript is one step away from the Cell paper by Winnubst et al (including here transcriptomics information). However, given the methodological feat here, this work is a huge achievement and the result of a large collaborative effort. The new insights however could be more

significant with a bit of effort from the authors. Clearly, the authors did not think that it is necessary in this work to analyze their data further (see their responses to the reviewers). More insights will most likely come from the subsequent usage of this dataset.

As stated above (and previously), within this single paper (which already contains a huge amount of content as a culmination of several years of work by nearly 80 co-authors), we are not able to add more analyses of our data at this time. But, we very much share the enthusiasm of the referee about the data as a rich resource and the potential insights that can be generated from deeper analyses either by ourselves or by the community. In fact, we have already started new analysis efforts and are coupling them with additional experiments to further investigate the relationship between transcriptomics and connectional specificity.